# HALT: Hallucination Assessment via Log-probs as Time series

## Abstract

Hallucinations remain a major obstacle for large language models (LLMs), especially in safety-critical domains. We present **HALT** (Hallucination Assessment via Log-probs as Time series), a lightweight hallucination detector that leverages only the top-20 token log-probabilities from LLM generations as a time series. HALT uses a gated recurrent unit model combined with entropy-based features to learn model calibration bias, providing an extremely efficient alternative to large encoders. Unlike white-box approaches, HALT does not require access to hidden states or attention maps, relying only on output log-probabilities. Unlike black-box approaches, it operates on log-probs rather than surface-form text, which enables stronger domain generalization and compatibility with proprietary LLMs without requiring access to internal weights. To benchmark performance, we introduce **HUB** (Hallucination detection Unified Benchmark), which consolidates prior datasets into ten capabilities covering both reasoning tasks (Algorithmic, Commonsense, Mathematical, Symbolic, Code Generation) and general-purpose skills (Chat, Data-to-Text, Question Answering, Summarization, World Knowledge). While being 30× smaller, HALT outperforms **Lettuce**, a fine-tuned `modernBERT-base` encoder, achieving a 60× speedup gain on HUB. HALT and HUB together establish an effective framework for hallucination detection across diverse LLM capabilities.

## 1 Introduction

Large Language Models (LLMs) have achieved remarkable progress in producing fluent and coherent text. Yet, they remain notoriously prone to hallucinations outputs that contain information which is verifiably false or unsupported. Such hallucinations can range from subtly incorrect facts to entirely fabricated references, thereby undermining user trust and limiting the deployment of LLMs in high-stakes applications. For instance, Li et al. (2023) reported that GPT-3.5 hallucinated in nearly 19.5% of user queries by introducing unverifiable details. The ability to reliably detect these errors is therefore essential for building trustworthy AI systems Ji et al. (2023).

A variety of approaches have been proposed for hallucination detection. Some assume white-box access to model internals Sriramanan et al. (2024); Chen et al. (2024b), an unrealistic requirement when dealing with closed-source APIs. Others rely on external retrieval augmentation Mishra et al. (2024); Friel et al. (2025) or additional API calls Manakul et al. (2023), both of which introduce latency and cost overheads. In many real-world scenarios, particularly with proprietary APIs, intermediate representations are inaccessible and even full output distributions may be hidden. Nonetheless, certain APIs for LLMs do expose limited metadata such as token-level log-probabilities. These values represent the model's confidence at each generation step and can serve as a lightweight signal for uncertainty.

This observation motivates our central research question: **Can hallucinations be detected by modeling only the sequence of token log-probabilities without analyzing the generated text itself or consulting external references?**

In principle, well-calibrated models assign higher probabilities to tokens consistent with training distributions Guo et al. (2017); Minderer et al. (2021), suggesting that high confidence might correlate with factual correctness. However, this assumption often breaks: (a) pretraining corpora may contain contradictory evidence in varying proportions, and (b) models may be poorly calibrated

such that predicted probabilities do not faithfully reflect correctness Desai & Durrett (2020). Thus, high probability is not reliable evidence of truthfulness. To address this, we propose a hallucination detector that treats token log-probabilities as a *time series*, classifying hallucinations based on their temporal dynamics rather than absolute values. Our method is strictly **black-box**: it does not access model weights, hidden states, retrieval systems, or surface-form text, thereby avoiding dependence on auxiliary LLMs that attempt to judge factual or logical consistency, an approach itself prone to hallucinations and domain biases Manakul et al. (2023). We hypothesize that while raw probabilities alone do not encode correctness, their evolving patterns provide stable and model-agnostic signals of uncertainty.

Unlike prior uncertainty-based methods that rely on aggregate statistics such as mean confidence or entropy Varshney et al. (2023); Quevedo et al. (2024), our approach leverages the entire ordered sequence of log-probabilities, capturing fluctuations across the generation process. A key advantage is its ability to naturally accommodate variable-length responses, since each output yields a log-probability trajectory of corresponding length. To model these trajectories, we draw from advances in time-series classification Ismail Fawaz et al. (2019), employing lightweight sequence models that label responses as hallucinated or not. This design adds negligible overhead, requiring only the log-probability stream that is often available during generation. Moreover, the method is fully model-agnostic: it can be applied to any LLM that exposes token likelihoods, making it particularly attractive for API-based deployments where injecting additional prompts or external verification checks is impractical.

In summary, our work makes the following three contributions:

- We introduce a new **black-box paradigm** for hallucination detection that relies solely on token log-probabilities. Our approach deliberately avoids surface-form text and external validators, making it applicable to closed-source APIs and robust against hallucinations in auxiliary models. Our framework **frames log-probability trajectories as a time-series classification problem**. Our lightweight model, **HALT**, is a 5M-parameter GRU Cho et al. (2014) that outperforms a fine-tuned ModernBERT encoder 30× larger. We release two HALT variants, HALT-L and HALT-Q, trained respectively on Llama 3.1-8B and Qwen 2.5-7B log-probabilities, demonstrating that compact sequence models can capture temporal uncertainty patterns overlooked by aggregate confidence metrics Varshney et al. (2023).

- We present **HUB (Hallucination detection Unified Benchmark)**, a benchmark spanning 10 LLM capabilities. HUB extends prior datasets such as FAVA Mishra et al. (2024), RAGTruth Niu et al. (2024) and HalluEval Li et al. (2023), leveraging CriticBench Lan et al. (2024) dataset by incorporating *logical hallucinations*, reasoning-related errors that move beyond factuality, opening the way for systematic study of both factual and logical hallucinations.

## 2 HUB: HALLUCINATION DETECTION UNIFIED BENCHMARK

### 2.1 SCOPE

To the best of our knowledge, no existing hallucination detection benchmark provides broad coverage across the full spectrum of large language model (LLM) capabilities. Prior efforts have instead concentrated on a narrow subset of tasks. For instance, **RAGTruth** Niu et al. (2024) is restricted to reference-based settings, covering only three capabilities: *Data-to-Text*, *Question Answering (QA)*, and *Summarization*. Similarly, **HaluEval** Li et al. (2023) targets the same three categories but substitutes *Data-to-Text* with *Dialogue*, emphasizing conversational skills. The annotated subset of **FAVA** Mishra et al. (2024) focuses largely on knowledge-intensive queries, with additional samples drawn from **OpenAssistant** Köpf et al. (2023) and **NoRobots** Rajani et al. (2023), though the scope remains limited.

In contrast, we expand beyond these resources by incorporating reasoning-focused tasks from **CriticBench** Lan et al. (2024), thereby constructing a more comprehensive benchmark that spans **ten LLM capabilities** essential for real-world applications. Specifically, HUB includes both:

- **Reasoning-oriented capabilities:** Algorithmic Reasoning, Commonsense Reasoning, Mathematical Reasoning, Symbolic Reasoning, and Code Generation.
- **General-purpose capabilities:** Chat, Data-to-Text, Question Answering, Summarization, and World Knowledge.

While prior work Mishra et al. (2024); Li et al. (2023); Niu et al. (2024) has focused mainly on knowledge-intensive or reference-grounded settings, we argue that incorrect outputs in reasoning tasks are also a form of hallucination. LLMs do not execute symbolic programs or arithmetic mechanistically; they generate plausible continuations of reasoning traces. When these traces yield invalid steps, inconsistent logic, or fictitious constructs, the model has effectively *hallucinated*. Thus, reasoning failures fall naturally under hallucination detection. They reflect the same lack of faithfulness observed in QA, summarization, or RAG, but along a logical rather than factual dimension. By adopting this broader view of *semantic faithfulness to the task specification*, HUB unifies factual and reasoning errors under a single benchmark.

To ensure generalization and prevent overfitting, each CriticBench capability cluster Lan et al. (2024) is built from multiple datasets, with one dataset per capability reserved for validation and the rest held out for testing. In CriticBench, Annotation reliability is maintained through a hybrid pipeline: rule-based heuristics, GPT-4-based annotation, and human adjudication whenever disagreements arise, balancing scalability with accuracy.

## 2.2 Splits and Generalization Protocol

As show in Table 1, the final HUB benchmark is divided into three splits: *train*, *validation*, and *test*. To rigorously assess generalization across capabilities, we deliberately restrict training to samples drawn from **Chat**, **Data-to-Text**, and **Question Answering**. These domains are sufficiently diverse to capture generic hallucination patterns while leaving other capabilities for out-of-distribution evaluation. Validation and test sets contain parallel samples from the same clusters to allow within-capability monitoring.

For external evaluation, we additionally incorporate human-annotated test sets from prior work. Specifically:

- A balanced subset of 500 examples from **HaluEval** Li et al. (2023) is held out for testing, while the remainder is split between training and validation.
- The human-annotated **FAVA Annotations** subset Mishra et al. (2024) is included as a gold-standard test set.
- The test portion of **RAGTruth** Niu et al. (2024) is also incorporated for testing.

This design yields a benchmark that not only spans a wide variety of LLM capabilities but also allows us to empirically validate whether our proposed method can approximate calibration biases, thereby enabling reliable hallucination detection across both in-domain and out-of-domain tasks.

## 2.3 Analysis

We analyze HUB in terms of class balance, capability coverage, and linguistic characteristics. This analysis highlights both the diversity of the benchmark and the challenges it poses for hallucination detection models.

Table 1 reports the distribution of samples across task clusters and dataset splits, together with the proportion of hallucination-labeled responses (shown in parentheses). Overall, HUB consists of **60,008** training samples (50.0% hallucinations), **7,342** validation samples (48.3% hallucinations), and **8,114** test samples (47.27% hallucinations). This near-balance across splits ensures fairness in training while preserving natural skew at the cluster level.

The hallucination ratio in HUB varies sharply across clusters: *World Knowledge* is heavily imbalanced ($\sim$95% hallucinations in validation, 80% in test), while clusters such as *Chat* and *Summarization* are closer to balanced ($\sim$40–50%). This variability makes **macro-averaged metrics** (e.g., macro-$F_1$) essential, since micro-averaging would be dominated by high-resource clusters like *Chat*, *QA*, or *Summarization*. Macro-averaging also ensures that underrepresented but critical capabilities

Table 1: Cluster-level dataset statistics. Each split is broken down into number of responses (Size), hallucination ratio (Ratio), and average response length in words (Len). Several clusters withheld from training to evaluate cross-task generalization (-)

| Task Cluster | Train | | Validation | | | Test | | |
|---|---|---|---|---|---|---|---|---|
| | Size | Ratio | Size | Ratio | Len | Size | Ratio | Len |
| Algorithmic | - | - | 32 | 50.00% | 29.97 | 250 | 32.00% | 55.35 |
| Chat | 11278 | 39.97% | 1991 | 39.98% | 35.28 | 1278 | 52.03% | 86.69 |
| Code Generation | - | - | 164 | 65.24% | 158.64 | 300 | 61.67% | 39.82 |
| Commonsense | - | - | 229 | 32.31% | 36.57 | 900 | 47.00% | 28.11 |
| Data2Text | 2759 | 50.02% | 487 | 49.90% | 157.83 | 900 | 64.33% | 156.70 |
| Mathematical | - | - | 300 | 46.67% | 42.03 | 1004 | 72.41% | 73.45 |
| QA | 35377 | 53.20% | 1885 | 50.34% | 35.19 | 1400 | 29.29% | 72.16 |
| Summarization | 10594 | 50.08% | 1870 | 50.05% | 73.78 | 1400 | 32.43% | 92.14 |
| Symbolic | - | - | 146 | 41.10% | 66.86 | 500 | 32.60% | 52.99 |
| World Knowledge | - | - | 238 | 94.96% | 139.03 | 182 | 80.22% | 246.02 |
| **Overall** | **60008** | **50.00%** | **7342** | **48.31%** | **-** | **8114** | **47.27%** | **-** |

(e.g., *Symbolic Reasoning*, *World Knowledge*) contribute equally while capturing errors from **both classes**: **false positives** (flagging faithful outputs) and **false negatives** (missing hallucinations).

Beyond class ratios, HUB displays substantial linguistic diversity due to spanning multiple task clusters with diverse response lengths: *World Knowledge* responses are longest (139–246 words), *Commonsense Reasoning* and *Algorithmic* are shortest (<40 words), *Summarization* is consistently verbose, and *Code Generation* remains compact. Overall, HUB embodies three properties: (i) highly imbalanced hallucination ratios, motivating macro-averaged evaluation; (ii) broad linguistic diversity, from terse algorithmic traces to verbose knowledge explanations; and (iii) heterogeneous coverage across splits, supporting both in-domain and cross-domain generalization. These make HUB both broad in scope and a challenging testbed for robust hallucination detection.

## 3 METHODOLOGY

### 3.1 MOTIVATION

Large Language Models (LLMs) differ in their *calibration*—the alignment between predicted token probabilities and actual correctness. Recent work has used *summary statistics* of token probabilities (e.g., mean confidence, entropy) as features for hallucination detection Sriramanan et al. (2024); Quevedo et al. (2024).

In this work, we extend this line of research by framing calibration as a *model-specific bias* and modeling it directly. Rather than collapsing probabilities into aggregate statistics, we represent the top-$k$ log probabilities at each decoding step as a rich time-series signal. We then train a gated recurrent unit (GRU) model to capture temporal patterns in this signal that reflect the model's calibration behavior.

Let $\mathcal{M}_\theta$ be an LLM with parameters $\theta$. During autoregressive generation, it outputs a distribution over the vocabulary at each step. Let $\boldsymbol{p}_t = \left(p_t^{(1)}, \ldots, p_t^{(k)}\right)$ be the top-$k$ probabilities at timestep $t$, where $k$ is fixed (e.g., $k = 20$ in our experiments). We define the log probability vector as:

$$\boldsymbol{\ell}_t = \left(\log p_t^{(1)}, \ldots, \log p_t^{(k)}\right) \in \mathbb{R}^k \tag{1}$$

A given LLM response with $T$ tokens can be summarized as $\boldsymbol{\ell}_{1:T} = (\boldsymbol{\ell}_1, \ldots, \boldsymbol{\ell}_T) \in \mathbb{R}^{T \times k}$.

*The top-k log-probability vectors capture the local structure of the model's predictive uncertainty, how sharply it scores the leading token relative to plausible alternatives. These patterns can be learned by a GRU to detect hallucinations.*

As an illustration for calibration bias, let $c_t \in \{0, 1\}$ indicate whether token $y_t$ is correct (i.e., faithful to reference or ground truth). A model is perfectly calibrated if:

$$\mathbb{P}(c_t = 1 \mid p_t^{(i)}) = p_t^{(i)} \quad \text{for } i \in \{1, \dots, k\}. \tag{2}$$

In practice, this equality rarely holds. We define the **calibration bias** function as:

$$b_\theta(p_t^{(i)}) = \mathbb{P}(c_t = 1 \mid p_t^{(i)}) - p_t^{(i)}. \tag{3}$$

**Hypothesis 1 (Model-Specific Bias).** For each LLM $\mathcal{M}_\theta$, there exists a deterministic function $b_\theta$ that governs the calibration behavior of top-$k$ token probabilities.

Each vector $\boldsymbol{\ell}_t$ contains the log-scale confidence over the top-$k$ tokens at time $t$, capturing both the sharpness of the distribution and how alternatives are scored. Over time, the sequence $\boldsymbol{\ell}_{1:T}$ might reveal patterns that help in detecting hallucinations.

**Hypothesis 2 (Bias Embedding and Learnability).** The sequence of top-$k$ log probability vectors $\boldsymbol{\ell}_{1:T}$ encodes the calibration bias function $b_\theta$. A GRU $f_\theta$ can learn an approximation of this bias-induced dynamics, enabling it to associate calibration patterns with hallucinations.

**Hypothesis 3 (Non-Transferability Across Models).** If $\mathcal{M}_\theta$ and $\mathcal{M}_{\theta'}$ are two different LLMs, then:

$$f_\theta(\boldsymbol{\ell}_{1:T}) \not\approx f_{\theta'}(\boldsymbol{\ell}_{1:T}),$$

since their calibration bias functions $b_\theta$ and $b_{\theta'}$ differ. Thus, a detector trained on one model does not transfer *reliably* to another.

**Hypothesis 4 (Task Generalization).** For a fixed LLM $\mathcal{M}_\theta$, a detector $f_\theta$ trained on hallucinations from task $\mathcal{T}_1$ generalizes to another task $\mathcal{T}_2$, because the underlying calibration bias $b_\theta$ is consistent across tasks.

We validate the above hypotheses empirically in Section 4.

## 3.2 APPROACH

**Feature Extraction.** Given annotated conversations where the final assistant turn is labeled as hallucinated or not, we extract token-level log-probability features using vLLM Kwon et al. (2023). We *teacher-force* the full conversation into the LLM $\mathcal{M}_\theta$, ensuring the gold response is generated token by token. At each step $t$, we record the top-20 log probabilities, motivated by Appendix D.2, which shows that $k = 20$ captures nearly the full predictive distribution and yields the strongest performance:

$$\boldsymbol{\ell}_t = (\log p_t^{(1)}, \dots, \log p_t^{(20)}) \in \mathbb{R}^{20}.$$

The first entry always corresponds to the *selected* token; if it is not the greedy choice, $\boldsymbol{\ell}_t$ contains the selected token followed by the top-19 alternatives. Thus each response of length $T$ becomes a sequence

$$\boldsymbol{\ell}_{1:T} \in \mathbb{R}^{T \times 20}.$$

For every $\boldsymbol{\ell}_t$, we additionally compute lightweight summary statistics capturing local calibration behavior, following Sriramanan et al. (2024); Quevedo et al. (2024), including entropy, selected-vs.-runner-up margin, and cumulative top-$k$ mass.

These token-level features are concatenated to $\boldsymbol{\ell}_t$, yielding an enriched feature vector:

$$\tilde{\boldsymbol{\ell}}_t = \left[ \phi(\boldsymbol{\ell}_t) \mid \boldsymbol{\ell}_t \right],$$

where $\phi(\boldsymbol{\ell}_t)$ denotes the vector of summary statistics.

The final input to our model is therefore a time series

$$\tilde{\boldsymbol{\ell}}_{1:T} \in \mathbb{R}^{T \times d},$$

where $d = d_{\text{stats}} + 20$ ($d_{\text{stats}} = 5$ in our experiments) combines raw log-probability features with engineered summary statistics.

**Comment.** This design directly follows from Hypotheses 1 and 2: the raw log-probability vectors $\boldsymbol{\ell}_t$ encode the calibration bias $b_\theta$, while the additional summary features highlight interpretable signals that have been shown useful in prior work Sriramanan et al. (2024); Quevedo et al. (2024).

**From top-**$20$ **log-probs to a proximal distribution.**    At each step $t$, we obtain a $k$-dimensional log-probability vector $\boldsymbol{\ell}_t = (\ell_t^{(0)}, \ldots, \ell_t^{(k-1)})$ with $k = 20$, where $\ell_t^{(0)}$ corresponds to the *selected* token and the remaining entries are the top-19 alternatives.[1] We convert these scores into a truncated, numerically stable probability distribution using

$$m_t = \max_i \ell_t^{(i)}, \qquad \tilde{p}_t^{(i)} = \frac{\exp(\ell_t^{(i)} - m_t)}{\sum_{j=0}^{k-1} \exp(\ell_t^{(j)} - m_t)}. \tag{4}$$

The resulting $\tilde{\boldsymbol{p}}_t \in \Delta^{k-1}$ is simply the model's predictive distribution *restricted, renormalized and near optimal D.2* over the top-$k$ support, preserving relative confidence among the most influential candidates without requiring access to the full vocabulary.

**Selected features.**    Let $\boldsymbol{\ell}_t$ and $\tilde{\boldsymbol{p}}_t$ be as above, and define $\text{alts} = \{1, \ldots, k-1\}$. We extract the following token-level features and feed their sequences to the GRU.

**1. Average log-probability**    This is a compact surrogate for *sharpness*: a more peaked local landscape (higher typicality) pushes the average log-probability upward (less negative), whereas a flatter/confused landscape (often preceding errors) lowers it. Averaging across the truncated support denoises single-token idiosyncrasies while staying sensitive to local certainty.

$$\text{AvgLogP}(t) = \frac{1}{k} \sum_{i=0}^{k-1} \ell_t^{(i)}. \tag{5}$$

**2. Rank proxy of the selected token**    Let $\ell_t^{(0)}$ be the selected token's log-prob. We define a bounded rank proxy within the top-20 window:

$$\text{RankProxy}(t) = 1 + \sum_{i \in \text{alts}} \mathbf{1}\big[\ell_t^{(i)} > \ell_t^{(0)}\big] \in \{1, \ldots, 20\}. \tag{6}$$

Lower values (near 1) indicate greedy selections, whereas higher values capture non-greedy or low-scoring selections. This feature directly quantifies *decision atypicality*, a known precursor of hallucinations when stochastic process of sampling selects a low confidence token.

**3. Overall entropy on the truncated (top-k) distribution**

$$H_{\text{overall}}(t) = - \sum_{i=0}^{k-1} \tilde{p}_t^{(i)} \log \tilde{p}_t^{(i)}. \tag{7}$$

This measures uncertainty over the *selected + alternatives* set. Elevated $H_{\text{overall}}$ flags indecision (many similarly likely candidates), whereas low entropy indicates a confident, peaked belief. Both abrupt spikes and collapses in $H_{\text{overall}}$ are informative dynamics around failure points.

**4. Alternatives-only entropy**    Let $\tilde{\boldsymbol{p}}_t^{\text{alts}}$ be $\tilde{\boldsymbol{p}}_t$ renormalized over the alternatives:

$$\tilde{p}_t^{\text{alts}}(i) = \frac{\tilde{p}_t^{(i)}}{\sum_{j \in \text{alts}} \tilde{p}_t^{(j)}} \quad (i \in \text{alts}), \tag{8}$$

$$H_{\text{alts}}(t) = - \sum_{i \in \text{alts}} \tilde{p}_t^{\text{alts}}(i) \log \tilde{p}_t^{\text{alts}}(i). \tag{9}$$

$H_{\text{alts}}$ isolates the *disagreement among competitors*: high values mean many plausible alternatives (ambiguous context), while low values mean a single strong challenger (knife-edge decisions). This complements $H_{\text{overall}}$ by probing the pressure the selected token faces.

---

[1] If the selected token is not the greedy choice, we include it plus the top-19 other candidates.

**5. Temporal change in binary decision**   Define the *binary* decision entropy between the selected token and the best alternative:

$$i_t^\star \;=\; \arg\max_{i\in\text{alts}} \ell_t^{(i)}, \tag{10}$$

$$p_c(t) \;=\; \frac{\exp(\ell_t^{(0)})}{\exp(\ell_t^{(0)}) + \exp(\ell_t^{(i_t^\star)})} \tag{11}$$

$$H_{\text{dec}}(t) \;=\; -\big[p_c(t)\log p_c(t) + (1 - p_c(t))\log(1 - p_c(t))\big]. \tag{12}$$

We use the *temporal delta* to capture sharp transitions:

$$\Delta H_{\text{dec}}(t) \;=\; H_{\text{dec}}(t) - H_{\text{dec}}(t-1)\,. \tag{13}$$

Positive jumps (*indecision spikes*) or negative drops (*snap-to-confident*) around critical steps are highly predictive signals for hallucination onsets or recoveries. *Implementation note:* even if $H_{\text{dec}}(t)$ is not appended as a feature, it must still be *computed* internally to make $\Delta H_{\text{dec}}(t)$ meaningful.

**6. Raw top-20 log-probabilities**   Finally, we pass the uncompressed vector $\ell_t$ itself. This exposes the GRU to the full *shape* of the local confidence landscape, including fine-grained margins and tail behavior that scalar summaries may miss. Empirically, retaining $\ell_t$ boosts robustness and lets the model discover interaction patterns (e.g., "one strong rival + many negligible tails") that are hard to hand-design.

**Features and Architecture.**   $\mathrm{AvgLogP}$ tracks distribution sharpness, $\mathrm{RankProxy}$ reflects non-greedy or atypical choices, $H_{\text{overall}}$ measures global uncertainty over the influential set, $H_{\text{alts}}$ captures dispersion among competitors, $\Delta H_{\text{dec}}$ detects rapid certainty–uncertainty transitions, and the raw $\ell_t$ retains high-resolution structure. Together, these features provide complementary *coarse* (entropy, average-based) and *fine-grained* (rank, local shape) views of calibration behavior, an observation confirmed by the attribution and ablation analyses in Appendix C, which show that HALT relies on the interplay of these signals rather than any single feature.

To model these signals, we use a bidirectional GRU encoder with a pooling head. Each response is represented as a sequence of token-level feature vectors (Sec.3.2), projected into a compact embedding space and processed by a multi-layer GRU. Variable-length sequences are aggregated using *Top-q pooling*, which averages the most salient timesteps (those with the largest hidden-state norms), emphasizing moments of sharp confidence shifts often diagnostic of hallucination. A final linear layer produces a single logit, trained with binary cross-entropy loss. Further details and ablations are in AppendixB.

## 4 RESULTS

We compare *white-box* baselines from LLMCheck Sriramanan et al. (2024) (requiring internal states), *aggregated-statistics* baselines (token-probability summaries), *black-box* text models, and our **HALT**. Unless noted otherwise, thresholds for sentence-level decisions are tuned on the HUB validation set and then held fixed for all test evaluations. From the token-level sequences, we reduce each metric to a single scalar per response: we take the *mean* over timesteps for all statistics, and the *maximum* for $\mathrm{RankProxy}$. A decision threshold for each metric is selected on the HUB validation set to maximize macro-$F_1$ and then applied to test sets (including FAVA and RAGTruth subsets). Lettuce Ádám Kovács & Recski (2025) predicts hallucination *spans* given the full conversation. We convert span outputs to a sentence label by marking a response as hallucinated if *any* span is predicted with probability $\geq 0.5$.

HUB clusters exhibit varying hallucination prevalence, from highly imbalanced settings (e.g., World Knowledge $\sim$95% hallucinations in validation) to more balanced ones (e.g., Chat, Summarization $\sim$40–50%). We therefore adopt **macro-$F_1$** as the primary metric, as it weights classes equally and avoids domination by skewed clusters. We additionally report AUROC (threshold-free discrimination) and standard $F_1$ for completeness. While both **FAVA annotations** Mishra et al. (2024) and **RAGTruth** Niu et al. (2024) are already part of HUB, we report their results separately in order to compare against prior published baselines that only evaluate on these subsets.

Table 2: Macro-$F_1$ scores on HUB test clusters. Aggregated statistics baselines are compared against the span-based Lettuce detector and our HALT variants. Summary stastistics are based on Llama 3.1 8B model log-probabilities. Best per-cluster scores are in **bold**, second best are underlined.

| Cluster | PPL | $H_{\text{overall}}$ | $\Delta H_{\text{dec}}$ | $H_{\text{alts}}$ | Lettuce | HALT-L | HALT-Q |
|---|---|---|---|---|---|---|---|
| Algorithmic | 24.24 | 24.24 | 26.44 | 24.48 | 24.24 | **76.80** | 32.68 |
| Chat | 35.09 | 34.55 | 37.41 | 36.47 | 41.50 | **60.17** | 58.60 |
| Code Generation | 43.07 | 38.14 | 62.03 | **66.67** | 38.14 | 47.67 | 39.71 |
| Commonsense | 33.25 | 31.97 | 35.17 | 34.27 | 41.06 | **56.67** | 41.32 |
| Data2Text | 39.15 | 39.15 | 42.18 | 39.15 | **83.38** | 72.89 | 73.00 |
| Mathematical | 44.14 | 42.00 | 66.95 | 61.63 | 41.80 | **72.71** | 62.90 |
| QA | 28.31 | 25.73 | 49.42 | 43.40 | **77.30** | 74.07 | 68.78 |
| Summarization | 24.49 | 24.49 | 45.03 | 32.99 | 59.71 | 66.93 | **70.75** |
| Symbolic | 24.59 | 24.59 | 24.92 | 24.59 | 33.36 | **65.40** | 49.78 |
| World Knowledge | 44.51 | 44.51 | 44.51 | 44.51 | 44.51 | **76.92** | 58.45 |
| Overall | 33.95 | 32.86 | 48.03 | 42.90 | 64.00 | **67.01** | 62.74 |
| Average | 34.08 | 32.94 | 43.41 | 40.81 | 48.50 | **63.03** | 55.60 |

Table 3: Comparison on the FAVA (left) and RAGTruth (right) subsets. Best scores per metric are in **bold**, second-best are underlined. Macro-$F_1$ is omitted (–) where not reported in prior baselines.

| Method | FAVA-Annotations | | | RAGTruth | | |
|---|---|---|---|---|---|---|
| | AUROC | F1 | Macro-$F_1$ | AUROC | F1 | Macro-$F_1$ |
| **White Box** | | | | | | |
| LLM-Check Attn Score | **68.19** | 70.53 | – | 58.30 | 57.18 | – |
| LLM-Check Hidden Score | 57.10 | 65.38 | – | 57.24 | 47.45 | – |
| **Aggregate Statistics** | | | | | | |
| $H_{\text{overall}}$ | 53.88 | 80.00 | 40.34 | 36.98 | 51.77 | 25.89 |
| $H_{\text{alts}}$ | 48.89 | 80.00 | 44.01 | 66.30 | 53.98 | 37.24 |
| $\Delta H_{\text{dec}}$ | 50.93 | 80.00 | 44.24 | 65.65 | 56.24 | 47.30 |
| PPL | 46.12 | 80.00 | 41.77 | 63.02 | 51.80 | 26.01 |
| **Black Box** | | | | | | |
| FAVA Model | 53.29 | 79.90 | – | – | – | – |
| Lettuce | 45.77 | **80.67** | 40.33 | **82.64** | **74.50** | **80.00** |
| HALT-L (Ours) | 61.30 | 77.86 | **61.57** | 70.65 | 59.00 | 65.70 |

Table 2 reports macro-$F_1$ across HUB clusters.

We compare aggregated statistics, the span-based Lettuce detector, and our two HALT variants (**HALT-L** trained on LLaMA 3.1-8B, **HALT-Q** trained on Qwen 2.5-7B), and refer the reader to Appendix Subsection D.1 for a detailed analysis of HALT's cross-model generalization and transferability across architectures and scales.

HALT-L's hyperparameters were tuned on HUB validation and then directly transferred to HALT-Q without re-tuning, which partly explains its lower overall performance. Across HUB, HALT achieves the best results on 7/10 clusters and leads both overall (67.00) and average (67.02) scores. Lettuce performs strongly on knowledge-heavy clusters such as Data2Text (83.38) and QA (77.30), but lags on reasoning tasks where sequence-level calibration cues are more predictive.

Among aggregated baselines, $H_{\text{alts}}$ peaks on Code Generation (66.67) and $\Delta H_{\text{dec}}$ is competitive on Mathematical Reasoning (66.95), though both fall short of HALT. Interestingly, HALT-Q shines on Summarization (70.75), while HALT-L dominates Algorithmic, Commonsense, Symbolic, and World Knowledge clusters, highlighting the model-specific nature of calibration dynamics.

Table 4: Transferability results across HUB clusters. Each row reports the *average* AUROC, Accuracy, and Macro-$F_1$ *across clusters*. Best results are in **bold**, second-best are underlined.

| Model | AUROC | Accuracy | Macro-$F_1$ |
|---|---|---|---|
| HALT-L | **70.02** | **67.02** | **63.04** |
| HALT-Q | 61.11 | 59.65 | 55.60 |
| HALT-L on Qwen LogProbs | 63.99 | 58.61 | 54.20 |
| HALT-Q on LLaMA LogProbs | 55.24 | 52.62 | 50.01 |
| Lettuce | 59.05 | 61.82 | 48.50 |
| Constant-Positive | 50.00 | 49.60 | 32.17 |
| Constant-Negative | 50.00 | 50.40 | 32.60 |
| Random Baseline | 49.67 | 49.42 | 47.93 |
| Weighted Random Baseline | 50.73 | 56.65 | 50.69 |

As shown in Table 3, On FAVA, HALT-L proves robust under class imbalance, while Lettuce shows instability when moving from F1 to Macro-$F_1$, reflecting its bias toward predicting hallucinations in a dataset with 67% positives. Macro-averaging therefore offers a fairer evaluation. The FAVA model itself is a fine-tuned LLaMA-7B paired with a retriever, trained solely for hallucination detection; although larger by roughly 1400$\times$, HALT-L remains highly competitive. On RAGTruth, Lettuce is stronger due to its training data being drawn directly from this benchmark, but HALT-L still outperforms aggregate statistics and white-box baselines despite no dataset-specific tuning.

On HUB transferability (Table 4), HALT-L achieves the best results across all metrics, with HALT-Q consistently second. This confirms **Hypothesis 1** and **Hypothesis 2**: model-specific calibration bias can be effectively captured when training and evaluation are aligned on the same LLM. Cross-model transfer (HALT-L$\rightarrow$Qwen, HALT-Q$\rightarrow$LLaMA) produces substantial drops, directly supporting **Hypothesis 3** that calibration dynamics are not reliably transferable across models. Importantly, both HALT variants outperform Lettuce and all randomized or trivial baselines (*Constant-Positive*, *Constant-Negative*, uniform and weighted random), which serve as lower bounds for detector performance. This supports **Hypothesis 4**: once a model's calibration bias is learned, the detector generalizes across task families more robustly than text-level heuristics or chance-level predictors.

## 5 CONCLUSION

We introduced **HALT**, a lightweight hallucination detector that models top-$k$ token log-probabilities as a time series, learning *model-specific* calibration dynamics with a compact GRU. To evaluate broadly, we released **HUB**, a benchmark spanning ten clusters across factual and reasoning tasks, with hybrid annotation and dataset splits that ensure reliable generalization. By treating reasoning failures as hallucinations alongside factual errors, HUB unifies semantic unfaithfulness across domains.

Experiments show HALT consistently outperforms probability summaries and rivals larger text encoders while being far smaller and faster, validating our hypotheses: calibration bias is model-specific and learnable, generalizes across tasks within a model, but transfers poorly across models. Beyond these results, HALT opens a new direction: treating log-probability trajectories as a time-series signal for LLM analysis. This perspective enables research into online hallucination detection during generation, calibration-aware decoding strategies tailored to specific models, and new ways to couple log-prob dynamics with retrieval or verifier signals. Extending this paradigm to multilingual and domain-specific settings could further expand its impact.

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

## A  RELATED WORK

As LLMs began to be used for open-domain tasks, the scale of the problem became more evident as these models can confidently assert falsehoods. The NLP community has addressed hallucination both by trying to reduce its occurrence during training Mohammadzadeh et al. (2025); Sun et al. (2022) and prompting strategies Xu & Ma (2025) and by developing techniques to detect hallucinated outputs post hoc Sriramanan et al. (2024); Ji et al. (2023). Our work focuses on the latter to detect hallucinations under the challenging constraint of black-box access.

### A.1  WHITE-BOX DETECTION METHODS:

With internal access to the LLM, a rich set of signals can be exploited to predict hallucinations. One line of work examines the model's own hidden representations or activations for telltale signs of falsehood. For example, Azaria & Mitchell (2023) showed that an LLM's internal state "knows when it's lying" by training a classifier on the model's intermediate layer embeddings for true and false outputs and achieved strong hallucination detection on a True/False QA task. INSIDE, a recent work by Chen et al. (2024a) detects hallucinations by sampling multiple responses for the same prompt and examining the internal states of the model. It computes the covariance matrix over hidden activations of these various responses and performs an eigen-decomposition of that covariance. Hallucinations are then inferred by a lack of self-consistency across these responses at a population level. In contrast, LLM-Check Sriramanan et al. (2024) is designed to assess whether a single fixed output response is hallucinated, avoiding multiple sample generation and focusing instead on features like hidden activations, attention maps, and output probabilities within that single response. These white-box methods deliver strong detection performance in controlled settings, but they depend on being able to instrument the target model, accessing hidden activations, model parameters,

or modifying internal computation, which is infeasible for proprietary models delivered via APIs. An alternative is to approximate internal states using a surrogate model Sriramanan et al. (2024), though that method imposes greater computational cost and may suffer from fidelity issues.

## A.2 BLACK-BOX METHODS

Perhaps, the most intriguing category of methods are those that treat LLM as a black box and do not require external ground truth. These methods probe the model behavior through prompting or multiple generations, often making the model "judge itself" in clever ways. A prime example is SelfCheckGPT Manakul et al. (2023), a zero-resource approach that leverages self-consistency. The idea is generating multiple stochastically sampled responses for a given prompt and then compare those responses to each other. If the model actually "knows" the correct answer (i.e. is not hallucinating), the responses should be consistent in the factual claims they make. Conversely, if the responses diverge or contradict each other on key facts, it's a strong indicator that the model is hallucinating and unsure – effectively, the truth is not stored in its knowledge and the outputs are guesses.

SelfCheckGPT was shown to outperform many baselines in detecting factual errors in passages about biography facts Manakul et al. (2023). However, its drawback is the need to generate 20 samples per prompt to get a reliable signal, which is expensive and not suitable for real-time applications. It also assumes that hallucination is relatively rare and that consistency across samples is a reliable proxy for truth, assumptions that may not hold in all tasks or domains.

Another black-box strategy is to employ prompt chaining or self-evaluation prompts Kamoi et al. (2024); Ren et al. (2023). Here, after the model produces an answer, one can query either the same model or a stronger model (like GPT-4) with a question like: "Is the above answer factually correct? If not, which parts are likely incorrect?" or "Critique the previous answer and identify any unsupported claims." This uses the model (or a second model) as a critic. Indeed, recent benchmarks like CriticBench Lan et al. (2024) explicitly evaluate LLMs on their ability to act as a critic of given responses. However, using a powerful model as a checker effectively outsources the problem to another LLM, which might not always be accessible or affordable. Furthermore, there is no guarantee that an LLM will accurately judge its own output – models can be evasive or overly lenient about their mistakes, especially if asked to critique themselves. Prompting strategies can be brittle: how the question is phrased or whether the model is instructed to be "truthful" can influence the outcome. And the judge models can also be prone to producing hallucinations.

A more structured prompting approach is embodied by metamorphic testing frameworks like MetaQA Yang et al. (2025b).Instead of directly asking the model to judge its answer, MetaQA generates one or more mutated prompts that should not change a truthful answer but might expose a hallucination. For example, it could add a detail to the question or rephrase it; if the model's answer to the mutated prompt is inconsistent with the original answer, that inconsistency flags a hallucination. This approach requires multiple query-response cycles (increasing cost) but cleverly avoids needing external data: it uses the model's own behavior under variations of the input as evidence. Our work shares a similar spirit of extracting maximum signal from the model itself under minimal additional assumptions. However, instead of requiring multiple queries or outputs, we focus on signals available from a single generation run – namely, the token probabilities.

## A.3 CONFIDENCE AND UNCERTAINTY SIGNALS

Several prior works have attempted to use the model's output probabilities or confidence scores as an indicator of hallucination. Indeed, if a model is properly calibrated, one might expect it to assign lower probability (higher uncertainty) to tokens that correspond to made-up information, compared to tokens that correspond to well-known facts. In practice, LLMs are not perfectly calibrated and can be overconfident in their false outputs Varshney et al. (2023).

Nonetheless, researchers have designed metrics based on probabilities or entropy to catch likely errors. One approach is to compute the perplexity of the output under the model itself or another model: a hallucinated passage might have higher perplexity (i.e. the model finds it "surprising") when evaluated with a strong language model as an evaluator. In Quevedo et al. (2024), a small set of features derived from token log-probabilities was used to train a simple binary classifier, yield-

ing state-of-the-art results on some hallucination benchmarks. Those features included aggregate statistics like the average log-probability of tokens in the output and the minimum token probability observed, as well as measures of how flat or peaked the distribution was at each step (e.g. the difference between the top-1 and top-5 token probabilities).

Similarly, Varshney et al. (2023) proposed to flag portions of text where the model's confidence was below a certain threshold and then verify those portions separately, effectively focusing on low-confidence segments as potential hallucinations.

Our work builds on the intuition that the model's time-series of confidence holds rich information, but we move beyond hand-crafted features or simple thresholds. Instead, we let a learned classifier inspect the entire sequence of log-probabilities. This way, patterns such as an abrupt drop in confidence at a certain point, or oscillations in probability (maybe indicating indecision), can be picked up automatically.

By using a time-series classifier Ismail Fawaz et al. (2019), our method can, for example, learn that a sequence with steadily high confidence except for one sharp dip (perhaps when the model "makes up" a specific name or number) is likely a hallucination.

Importantly, this approach does not require any second model or external knowledge – it uses only the data from the model's single forward pass. Compared to multi-sample methods like SelfCheck-GPT, it is much more efficient (no need for 20 runs; just one run with minimal overhead).

Compared to prompting-based judges, it does not require an extra API call to another model or the same model in judge mode. And compared to static feature approaches Quevedo et al. (2024), it leverages the shape of the proximal probability curve -since most APIs return at most the top-20 log-probabilities for each token- rather than collapsing it to a few summary statistics, which we find improves detection performance.

## A.4 TIME-SERIES CLASSIFICATION PERSPECTIVE

Casting the detection problem as time-series classification also connects our work to a broad literature in sequence analysis. Techniques such as recurrent neural networks, 1-D convolutional networks, and transformer encoders have been widely used to classify time-series data (e.g. sensor readings, speech signals) of varying lengths Ismail Fawaz et al. (2019).

We adopt similar techniques here. In essence, our classifier can be seen as a small GRU Cho et al. (2014) that "reads" the sequence of $\log P(\text{token}|\text{context})$ values and outputs a label. This is analogous to sequence classification in NLP (like classifying a sentence as positive/negative sentiment, except here the "sentence" is a sequence of probability values rather than word embeddings). By leveraging this mature area of research, we ensure our model can handle different sequence lengths and learn temporal patterns effectively.

Previous works have not explicitly applied time-series modeling to sequences of model confidences for hallucination detection, which is the gap our work fills.

In summary, existing hallucination detection methods either use substantial external information (knowledge or multiple model outputs) or internal access to the model, or they simplify the confidence signals to a few features. Our approach is positioned at a unique point in this design space: it assumes only that we can obtain the model's token-level log probabilities – a reasonable capability for many modern LLM APIs or open-source models – and nothing else. Within this constraint, it uses a powerful sequence modeling approach to capture subtle signs of hallucination.

To the best of our knowledge, no prior work has utilized the full log-probability sequence in this manner. By doing so, we show that hallucination detection is possible even in the most restrictive deployment scenarios, and we provide a method that is complementary to more resource-heavy techniques. Our results (Section 4) will illustrate that this minimalist approach can nonetheless achieve competitive accuracy, highlighting an interesting and practical direction for safe LLM usage.

## B  ARCHITECTURE

**Overview.**  Given a token sequence of feature vectors $\tilde{\ell}_{1:T} \in \mathbb{R}^{T \times d}$ (Sec. 3.2), we employ a gated recurrent unit (GRU) encoder followed by a sequence-to-scalar pooling head and a linear classifier. The model predicts a sentence-level hallucination score (logit), later passed through a sigmoid during evaluation.

**Input projection and normalization.**  We first apply LayerNorm to each feature vector, then project to a lower-dimensional embedding using a two-layer MLP with GELU:

$$\tilde{\ell}_t \to \mathrm{LN}(\tilde{\ell}_t) \to \mathrm{MLP}_{d \to \mathrm{proj\_dim}} \quad (\mathrm{proj\_dim} = 128).$$

This stabilizes training and provides a compact representation when raw log-probabilities are appended.

**Bidirectional GRU encoder.**  The projected sequence is encoded by a multi-layer, bidirectional GRU:

$$\mathrm{GRU}(\cdot;\ \mathrm{hidden\_dim} = 256,\ \mathrm{num\_layers} = 5,\ \mathrm{bidirectional} = \mathrm{true},\ \mathrm{dropout} = 0.4).$$

We use `pack_padded_sequence`/`pad_packed_sequence` together with a boolean mask to handle variable-length responses efficiently and to ensure padded positions do not influence the hidden dynamics. The bidirectional configuration allows HALT to model both left-to-right and right-to-left uncertainty flows, important because many hallucination signatures (e.g., abrupt entropy spikes or sudden rank inversions) are better captured when the model observes temporal context from both directions.

### GRU VS. LSTM VS. RNN: EMPIRICAL COMPARISON

To validate the architectural choice, we trained three recurrent architectures (GRU, LSTM, Vanilla RNN) using the same training protocol and LLaMA-3.1-8B log-probabilities as input. Table 5 summarizes results across the HUB benchmark.

| Architecture | Overall F1 | Average F1 |
|---|---|---|
| **GRU** | **0.6701** | **0.6303** |
| LSTM | 0.6556 | 0.5919 |
| RNN | 0.5516 | 0.5072 |

Table 5: Comparison of recurrent encoders trained on LLaMA-3.1-8B log-probabilities. GRU outperforms both LSTM and RNN across HUB.

**Findings.**  The performance hierarchy is consistent across all HUB capability clusters:

- **GRU achieves the best results** in both overall and average macro-F1.
- LSTM performs competitively but worse than GRU, likely due to over-parameterization for this mid-size (25-dim) feature space and sequence lengths of 20–150 tokens.
- Vanilla RNN significantly underperforms, confirming that nonlinear gating is essential for modeling uncertainty trajectories.

These results reinforce our architectural choice: GRUs provide the right balance of expressiveness, temporal gating, and computational efficiency. Coupled with the bidirectional configuration and uncertainty-based features, they capture hallucination-relevant temporal dynamics more effectively than alternative recurrent architectures.

**Salient-timestep pooling (Top-$q$).**  Let $H \in \mathbb{R}^{B \times T \times D}$ be the GRU outputs (with $D = 2 \times$ hidden_dim due to bidirectionality). We score each timestep by its $\ell_2$ norm, mask out padding, and average the top-$q$ fraction per sequence ($q = 0.15$):

$$\mathrm{score}_t = \|H_t\|_2, \qquad \mathrm{pooled} = \frac{1}{K} \sum_{t \in \mathrm{Top\text{-}}q} H_t.$$

Top-$q$ pooling focuses the classifier on the most informative regions (e.g., bursts of uncertainty or sharp confidence shifts) instead of diluting signals over all tokens. We found it more robust than mean/max pooling and simpler than attention while retaining strong performance.

**Classification head.** The pooled vector optionally passes through a LayerNorm (disabled in our best setting, `out_norm=false`) and a linear layer to produce a single logit:

$$\hat{z} = \mathbf{w}^\top \text{pooled} + b, \qquad \hat{y} = \sigma(\hat{z}).$$

At training time we use `BCEWithLogitsLoss`, which combines the sigmoid and binary cross-entropy in a numerically stable way.

**Regularization and stability.** We employ dropout within the GRU stack (`dropout=0.4` between recurrent layers), LayerNorm on inputs, and gradient clipping (`max_norm=1.0`). These control overfitting and stabilize optimization when raw log-probabilities are included.

**Optimization (brief).** We train with Adam (`lr=`$4.41 \times 10^{-4}$, `weight_decay=`$2.34 \times 10^{-6}$), batch size 512, for up to 100 epochs, using `ReduceLROnPlateau` (factor 0.5, patience 3, mode=max) and early stopping (patience 15) on the validation metric. This schedule adapts the learning rate to plateauing validation performance and avoids overfitting while converging reliably.

**Design rationale.** (i) A bidirectional GRU captures temporal patterns in the confidence landscape without imposing strong parametric assumptions. (ii) Top-$q$ pooling concentrates on salient segments (e.g., spikes in decision entropy delta or sustained high alternative entropy) that are most diagnostic of hallucination. (iii) Input projection and LayerNorm make the model tolerant to heterogeneous feature scales when combining raw log-probs with summary features.

## C  A CLOSER LOOK INTO FEATURES

This appendix provides a detailed examination of the input features used by HALT and explains how the model leverages temporal uncertainty patterns to detect hallucinations.

HALT operates on a sequence of *25 features per timestep*, consisting of the top–20 token log-probabilities returned by the LLM and five engineered uncertainty features. To better understand their contributions, we performed two complementary analyses:

1. Gradient $\times$ Input attribution over all features.
2. Feature ablation across all ten HUB capability clusters.

Together, these experiments reveal that HALT relies on rich temporal dynamics rather than any single feature or static threshold.

### C.1  INPUT FEATURE SET

For each generated token, HALT receives a feature vector of dimension $F = 25$, composed of:

- **Top–20 log-probabilities**
- **Five engineered uncertainty features:**
  - `entropy_overall`
  - `entropy_alts`
  - `avg_logprob`
  - `rank_proxy`
  - `dec_entropy_delta`

These features allow HALT to observe both the *shape* and the *temporal evolution* of the LLM's predictive distribution.

## C.2 GRADIENT × INPUT ATTRIBUTION

To quantify feature importance, we compute gradient × input contributions for each feature across the full evaluation set. For an input tensor $x \in \mathbb{R}^{B \times T \times F}$ with corresponding gradients $g$, we estimate contribution as:

$$C = |g \odot x|,$$

followed by masking padded timesteps and summing contributions across batches and time.

Below is the core code fragment used in our analysis (included here for reproducibility):

```
contrib = (grads * x).abs()        # (B, T, F) gradient × input magnitude
mask = _make_mask(lengths, T, device=device)
contrib = contrib * mask.unsqueeze(-1)

feat_imp_batch = contrib.sum(dim=(0, 1))        # (F,)
time_imp_batch = contrib.sum(dim=2).sum(dim=0)  # (T,)
```

### C.2.1 GLOBAL FEATURE IMPORTANCE

Table 6 reports normalized importance weights for all 25 features.

| Feature | Importance |
|---------|------------|
| logprob_15 | 0.1138 |
| logprob_4 | 0.0818 |
| logprob_17 | 0.0698 |
| logprob_20 | 0.0580 |
| logprob_13 | 0.0557 |
| logprob_6 | 0.0553 |
| logprob_3 | 0.0539 |
| logprob_19 | 0.0532 |
| logprob_1 | 0.0426 |
| entropy_alts | 0.0419 |
| logprob_12 | 0.0370 |
| logprob_14 | 0.0343 |
| logprob_16 | 0.0336 |
| logprob_2 | 0.0333 |
| avg_logprob | 0.0317 |
| logprob_11 | 0.0314 |
| logprob_10 | 0.0308 |
| logprob_5 | 0.0260 |
| logprob_18 | 0.0243 |
| rank_proxy | 0.0226 |
| logprob_7 | 0.0220 |
| logprob_9 | 0.0172 |
| logprob_8 | 0.0144 |
| entropy_overall | 0.0116 |
| dec_entropy_delta | 0.0037 |

Table 6: Normalized global feature importance from gradient × input attribution.

**Interpretation.** The results show:

- HALT draws on a *mixture of signals*: several top–$k$ log-probabilities dominate, but engineered features (e.g., `entropy_alts`, `avg_logprob`, `rank_proxy`) also contribute substantially.
- `logprob_1` (the log-probability of the sampled token) is *not* the most influential feature, indicating that HALT does not merely rely on the likelihood of the generated token. Instead, it learns more structured temporal behaviors related to distributional uncertainty.

## C.3 Feature Ablation Across HUB Clusters

We also retrained HALT after removing each engineered feature individually, and evaluated performance across all ten HUB capability clusters using LLaMA-3.1-8B log-probabilities.

| Model Variant | Avg F1 | Overall |
|---|---|---|
| **full** | **0.630** | **0.670** |
| w/o avg_logprob | 0.600 | 0.657 |
| w/o entropy_overall | 0.598 | 0.665 |
| w/o rank_proxy | 0.595 | 0.654 |
| w/o dec_entropy_delta | 0.574 | 0.646 |
| w/o entropy_alts | 0.568 | 0.647 |

Table 7: Feature ablation results averaged over the ten HUB clusters.

**Findings.**

- Removing any engineered uncertainty feature leads to a *consistent* drop in performance.
- The largest degradations arise from removing entropy-based features, supporting the intuition that entropy spikes and instability in alternative-token probabilities are key hallucination indicators.
- These ablations corroborate the attribution analysis, demonstrating tight alignment between gradient-based interpretation and empirical contributions.

## C.4 Takeaway

The combined attribution and ablation analyses clarify which temporal signals HALT uses to detect hallucinations. The model's behavior is driven by:

- abrupt changes in high-rank log-probabilities,
- fluctuations in entropy over alternative tokens,
- shifts in rank proxies and average log-probabilities,
- concentration of contributions around "reasoning forks," where LLM uncertainty spikes.

These results demonstrate that HALT captures *interpretable temporal uncertainty dynamics*, directly addressing the reviewer's concern that the influential features were previously unclear. We will incorporate these findings into the main paper to strengthen the interpretability of HALT.

# D Additional Analyses on Model Generalization and Feature Design

## D.1 Cross-Model Generalization: Does HALT Transfer Across LLMs?

A natural question can be raised is whether HALT, when trained on the temporal uncertainty patterns of one model (e.g., LLaMA-3.1-8B), will generalize to others with different sizes, architectures, and calibration characteristics. To answer this, we expanded HALT training and evaluation across eight language models ranging from 360M to 70B parameters.

### Expanded Cross-Model Experiments

We trained HALT independently on each model's log-probability sequences without any modification to the HALT architecture. Except for HALT-L, which uses tuned hyperparameters, all other models were trained with the same configuration. For Qwen-7B, a light sweep improved performance from $0.62 \rightarrow 0.65$ (overall F1) and $0.55 \rightarrow 0.57$ (average F1).

Table 8 summarizes the results.

| Model | Params | Overall F1 | Average F1 |
|---|---|---|---|
| SmolLM Allal et al. (2024) | 360M | 0.5930 | 0.5265 |
| SmolLM Allal et al. (2024) | 1.7B | 0.6090 | 0.5390 |
| LLaMA 3.2 Grattafiori et al. (2024) | 3B | 0.6283 | 0.5601 |
| HALT-Q (Qwen 2.5) | 7B | 0.6274 | 0.5560 |
| **HALT-L (LLaMA 3.1)** | **8B** | **0.6701** | **0.6303** |
| Qwen 3 Yang et al. (2025a) | 14B | 0.6248 | 0.5264 |
| Qwen 3 Yang et al. (2025a) | 32B | 0.6406 | 0.5549 |
| LLaMA 3.1 Grattafiori et al. (2024) | 70B | 0.6592 | 0.5954 |

Table 8: HALT trained on log-probabilities from eight different models, ranging from 360M to 70B parameters.

**Findings.** The results reveal two important observations:

- **HALT transfers across architectures and scales.** Even without tuning, HALT achieves stable performance on models with very different internal calibration properties (e.g., LLaMA vs. Qwen vs. SmolLM).

- **Performance improves smoothly with model size, but not monotonically.** Smaller models (360M–1.7B) have flatter predictive distributions, making hallucination patterns noisier and harder to learn, yet HALT still performs reliably. Larger models (32B–70B) exhibit clearer uncertainty dynamics, yielding stronger results.

INTERPRETATION: WHY DOES HALT GENERALIZE?

The transferability of HALT across diverse LLMs is consistent with our attribution and top-$k$ analysis D.2:

- The temporal dynamics of entropy, rank shifts, and alternative-token interactions appear highly *model-agnostic*.

- Even when the absolute calibration differs substantially (e.g., SmolLM 360M vs. LLaMA 70B), the *patterns surrounding hallucination events* remain similar.

- HALT learns these patterns rather than memorizing model-specific logits.

This supports the broader claim that hallucination signatures are reflected in universal uncertainty trajectories, not model-specific probability scales.

CONCLUSION

These results indicate that HALT is robust across a wide range of LLM families and parameter counts. While tuning can yield modest improvements (as shown with HALT-Q), HALT's strong out-of-the-box performance demonstrates that:

> *HALT is not tied to any single model: it learns transferable uncertainty dynamics that generalize across architectures, sizes, and calibration regimes.*

This directly addresses the reviewer's concern and strengthens the case for HALT as a model-agnostic hallucination detector.

D.2   ON THE SIGNIFICANCE OF THE TOP-$k$ PARAMETER

HALT relies on the top-$k$ log-probabilities of the next-token distribution to characterize uncertainty dynamics. The choice of $k$ directly affects (i) how much of the predictive distribution HALT observes, and therefore (ii) the richness of the temporal signals available to the GRU. We conducted two analyses to understand this design choice: (1) varying $k$ during HALT training, and (2) estimating how much probability mass is captured by different $k$ values across diverse LLMs.

EFFECT OF $k$ ON HALT PERFORMANCE

We trained HALT-L (using LLaMA-3.1-8B log-probabilities) with $k \in \{1, 5, 10, 15, 20\}$. Table 9 summarizes the results.

| $k$ | Overall F1 | Average F1 |
|---|---|---|
| 1 | 0.5927 | 0.5464 |
| 5 | 0.6352 | 0.5563 |
| 10 | 0.6581 | 0.6043 |
| 15 | 0.6578 | 0.5816 |
| **20** | **0.6701** | **0.6303** |

Table 9: Impact of top-$k$ size on HALT-L performance.

**Findings.**

- Performance improves steadily as $k$ increases from 1 to 10, indicating that richer uncertainty information leads to better hallucination detection.
- The best results occur at $k = 20$, which provides a strong trade-off between informativeness and computational cost.
- Beyond $k = 10$, gains come primarily from capturing lower-ranked alternatives that exhibit distinctive temporal patterns around hallucination boundaries (e.g., entropy surges, sudden rank reversals).

These results align with the feature attribution and ablation analyses: HALT benefits from a diverse set of temporal signals, not just the sampled token's log-probability but also interactions among alternative probabilities.

HOW MUCH OF THE DISTRIBUTION DOES TOP-$k$ CAPTURE?

To evaluate whether top-$k$ is sufficient for entropy-based features, we measured the cumulative probability mass captured by top-$k$ across 12 LLMs on the HUB validation set. The results are strikingly consistent:

- Top-1 captures only 57–65% of mass.
- Top-5 captures 88–98%.
- Top-10 captures 93–99%.
- Top-15 captures 94–99%.
- Top-20 captures **95–99.7%**.

**Implications.**

- Entropy and rank-based signals are effectively determined by the top-20 portion of the distribution; contributions from the long tail are negligible.
- Increasing $k$ beyond 20 would have minimal impact but increase computational/storage overhead.
- Smaller models (e.g., Smol-LM 360M) exhibit flatter distributions and thus benefit disproportionately from larger $k$, whereas larger models (e.g., Smol-LM 1.7B) already concentrate probability mass and show smaller marginal gains.

CONCLUSION

The combined analyses show that:

1. Top-20 captures nearly the *entire effective distribution* relevant for uncertainty modeling.

2. HALT's performance is monotonically increasing with $k$ and peaks at $k = 20$ under computational constraints.

3. Larger $k$ values provide diminishing returns because the remaining probability mass is negligible and rarely influences entropy dynamics.

Thus, $k = 20$ is a principled choice that balances computational efficiency with maximally informative uncertainty features for hallucination detection.

