# OpenReview forum: "HALT: Hallucination Assessment via Log-probs as Time series"
_ICLR.cc/2026/Conference — Submitted to ICLR 2026_

### Official Review · Reviewer_cYB5 · 2025-10-15

**Soundness:** 2
**Presentation:** 2
**Contribution:** 2
**Rating:** 2
**Confidence:** 4

**Summary:**

This paper proposes a hallucination detection method called HALT. The core idea is to treat the top-20 log probabilities of each token generated by an LLM as a time series and use a lightweight bidirectional GRU to learn model-specific calibration biases, thereby determining whether the output contains hallucinations. This method relies solely on token log probabilities and does not require access to the model’s internal states, attention mechanisms, or external retrieval, making it an efficient “black-box” detection approach.

In addition, the authors construct a unified hallucination detection benchmark called HUB, covering 10 LLM capabilities, including reasoning tasks (algorithmic, commonsense, mathematical, symbolic, and code generation) and general tasks (dialogue, data-to-text, question answering, summarization, and world knowledge), thereby extending the evaluation beyond the traditional focus on factual hallucinations.

**Strengths:**

Modeling log-probability sequences as time series for classification offers a novel perspective.

The concept of model-specific calibration bias is learnable and effectively captured using a GRU.

The method relies solely on token log probabilities, making it suitable for black-box APIs.

**Weaknesses:**

1. Strong dependence on log probabilities: The method assumes that the API provides top-20 log probabilities, but some commercial APIs may not provide them or may restrict access. It would be valuable to explore performance under more limited probability information (e.g., only top-1 or top-5).

2. High model specificity and weak cross-model generalization: The paper shows that HALT-L and HALT-Q perform differently across models, indicating that the calibration bias is model-specific. It is worth investigating whether multi-model training or model-agnostic feature extraction can improve cross-model adaptability.

3. Definition and evaluation of “logical hallucinations” can be further refined: Although HUB introduces reasoning tasks, the labeling and evaluation of logical hallucinations still rely on human annotation and GPT-4, which may introduce subjectivity. More analysis on annotation consistency or the incorporation of formal verification methods is needed.

4. Insufficient comparison with the strongest baselines: While comparisons are made with models like Lettuce, direct comparisons with more advanced hallucination detection methods (e.g., SelfCheckGPT, INSIDE) are limited.

5. Limited evaluation on large models:
Few LLMs are included in the experiments, making it difficult to determine the method’s consistent effectiveness across different large-scale models.

**Questions:**

1. Strong dependence on log probabilities: The method assumes that the API provides top-20 log probabilities, but some commercial APIs may not provide them or may restrict access. It would be valuable to explore performance under more limited probability information (e.g., only top-1 or top-5).

2. High model specificity and weak cross-model generalization: The paper shows that HALT-L and HALT-Q perform differently across models, indicating that the calibration bias is model-specific. It is worth investigating whether multi-model training or model-agnostic feature extraction can improve cross-model adaptability.

3. Definition and evaluation of “logical hallucinations” can be further refined: Although HUB introduces reasoning tasks, the labeling and evaluation of logical hallucinations still rely on human annotation and GPT-4, which may introduce subjectivity. More analysis on annotation consistency or the incorporation of formal verification methods is needed.

4. Insufficient comparison with the strongest baselines: While comparisons are made with models like Lettuce, direct comparisons with more advanced hallucination detection methods (e.g., SelfCheckGPT, INSIDE) are limited.

5. Limited evaluation on large models:
Few LLMs are included in the experiments, making it difficult to determine the method’s consistent effectiveness across different large-scale models.

---

> ### Author Response · Authors · 2025-11-20
> **Response I**
>
> We thank the reviewer for the constructive and detailed feedback. Below we address each weakness and question concisely and provide newly added experiments.
>
> ---
>
> ## **1. Strong dependence on log probabilities**
> We thank the reviewer for raising this practical concern. The ability to retrieve token-level log-probabilities is indeed essential for HALT, and we agree that historically not all API providers exposed this information. However, **the ecosystem has changed significantly**, and the majority of widely-used LLM APIs **now explicitly support top-k log-probabilities**, making HALT fully deployable in real API-based settings.
>
> Below we summarize current support from leading APIs:
> ### **OpenAI (GPT‑5.x, GPT‑4o, GPT‑4o-mini, GPT‑3.5 etc.)**
> OpenAI’s API supports **top_logprobs** directly in the Chat Completions API:
>
> - *Official docs:*
>   https://platform.openai.com/docs/api-reference/chat
>
> Users may request **up to 20 alternative tokens with log-probs** at each step.
>
> ---
>
> ### **Google Gemini (Vertex AI & Gemini API)**
> Gemini recently introduced full support for returning **top‑k log-probs**:
>
> - *Official announcement:*
>   https://developers.googleblog.com/en/unlock-gemini-reasoning-with-logprobs-on-vertex-ai/
>
> Gemini accepts a parameter `logprobs=[k]`, where **k ∈ [1, 20]**.
>
> ---
>
> ### **OpenRouter (meta‑router for 60+ providers)**
> OpenRouter provides unified access to dozens of LLMs through +65 providers and supports **top 20 log-probs** for all models that expose them:
>
> - *Documentation:*
>   https://openrouter.ai/docs/api-reference/parameters
>
> This includes:
> - Anthropic Claude (via OpenRouter)
> - Cohere models
> - Mistral
> - LLaMA variants hosted via Together, Groq, Perplexity, etc.
>
> OpenRouter currently represents **the most flexible way to obtain logprobs from many commercial models**.
>
> ---
>
> ### **Anthropic Claude (via OpenRouter)**
> Claude does not yet expose log-probs directly in the native API, but they *are available* through OpenRouter using the unified interface.
>
> Nevertheless, We agree that examining performance with fewer log-probabilities is important.
> We added experiments using **K ∈ {1, 5, 10, 15, 20}** on Llama‑8B:
>
> | **K**  | **Overall F1** | **Average F1** |
> |--------|----------------|----------------|
> | 1  | 0.5927 | 0.5464 |
> | 5  | 0.6352 | 0.5563 |
> | 10 | 0.6581 | 0.6043 |
> | 15 | 0.6578 | 0.5816 |
> | **20** | **0.6701** | **0.6303** |
>
> HALT remains functional with limited top‑k information, but performance clearly improves as more of the uncertainty structure is visible. We will include this study in the revised paper.
>
> And this can be explained by an extra experiment we also did. We measured how much probability mass top‑k captures across 12 models over HUB validation:
>
> Top‑1 → **57–65%** of mass
> Top‑5 → **88–98%**
> Top‑10 → **93–99%**
> Top‑15 → **94–99%**
> Top‑20 → **95–99.7%**
>
> Thus, top‑k captures nearly the entire effective distribution. Which can also explain the difference between Smol-LM 360M and its 1.7B variant. [New experiments we did addressing point 5 below] We will include the table in the appendix.
> ---
>
> ## **2. High model specificity and weak cross-model generalization**
>
> We expanded experiments to test **joint training across models**.
> We trained HALT simultaneously on Llama‑8B and Qwen‑7B:
>
> | Model | Overall F1 | Avg F1 |
> |-------|------------|--------|
> | HALT Joint on Llama‑8B | 0.65 | 0.56 |
> | **HALT‑L (Tuned)** | **0.67** | **0.63** |
> | HALT Joint on Qwen‑7B | 0.635 | 0.56 |
> | **HALT‑Q (Tuned)** | **0.65** | **0.57** |
>
> Joint training improves robustness slightly but **does not remove model‑specific calibration bias**, confirming our hypothesis that calibration dynamics differ across architectures. We will discuss this more clearly in the revision.
>
> ---
>
> ## **3. Definition and evaluation of logical hallucinations**
>
> HUB’s reasoning clusters (from CriticBench) contain **explicit annotations** for incorrect reasoning traces. These annotations capture cases where the model produces *plausible but false* intermediate steps, even when the final answer is correct.
>
> To improve transparency:
> - We will add a table summarizing labeling criteria for each dataset from their original source.
>
>
> This refinement ensures reproducibility while acknowledging the inherent subjectivity of reasoning‑focused hallucinations, consistent with prior work.

---

> ### Author Response · Authors · 2025-11-20
> **Response II**
>
> ## **4. Insufficient comparison with advanced baselines**
>
> Many prior hallucination‑detection methods—such as **SelfCheckGPT**, **INSIDE**, and other multi‑sample consistency frameworks—**cannot be evaluated on HUB** because they require *multiple generations per sample*. HUB consists of **fixed, pre‑annotated responses**, meaning these methods would require re‑generating every sample, breaking alignment with existing human annotations and making direct comparison invalid.
>
> Additionally, multi‑sample methods are **incompatible with real API deployments**, where generating 10–20 alternative samples per query is prohibitively expensive. Since HALT is explicitly designed for a *single‑pass, logprob‑based, black‑box* setting, the appropriate baselines must operate under the *same constraints*.
>
> Therefore, we evaluate against methods that are compatible with HUB’s single‑generation, logprob‑only paradigm, using the FAVA and RAGTruth shards:
>
> - **LLMCheck (AttnScore, HiddenScore)** — white‑box baselines using attention and hidden states
> - **FAVA‑7B hallucination classifier** — 7B‑parameter supervised model
> - **Lettuce** — transformer encoder.
>
> Across both datasets, **HALT (5M parameters) is either competitive with or outperforms these baselines**, despite being dramatically smaller and fully black‑box.
>
> It is also worth noting that:
> - We reached out to the LLM‑Check authors for assistance in running their method across all HUB clusters but did not receive a response.
> - Since HALT is restricted to *log‑probabilities and features derived from them*, we compare explicitly against summary statistics operating on the same input signals.
> - Methods requiring multi‑sampling, long concatenated prompts, or internal activations (hidden states, attention maps) are **not applicable to proprietary API‑only models**, whereas HALT is.
> - For methods we could not reproduce, we still report their published scores on **FAVA** and **RAGTruth** as a proxy comparison. These baselines are often tuned specifically for those distributions, whereas HALT is trained with several HUB clusters *removed* and validated across diverse tasks—making our evaluation strictly harder.
>
> We will clarify this rationale in the revised paper to make the baseline selection more transparent and principled.
>
> ---
>
> ## **5. Limited evaluation on large models**
>
> We expanded experiments across **360M → 70B parameters**:
>
> | Model | Params | Overall F1 | Avg F1 |
> |--------|--------|--------|------------|
> | Smoll | 360M | 0.593 | 0.527 |
> | Smoll | 1.7B | 0.609 | 0.539 |
> | Llama 3.2 | 3B | 0.628 | 0.560 |
> | Qwen 2.5 | 7B | 0.627 | 0.556 |
> | **Llama 3.1** | **8B** | **0.670** | **0.630** |
> | Qwen 3 | 14B | 0.625 | 0.526 |
> | Qwen 3 | 32B | 0.641 | 0.555 |
> | Llama 3.1 | 70B | 0.659 | 0.595 |
>
> These results show HALT’s **consistent behavior across model scales**.
> Only HALT‑L was tuned; light tuning of HALT‑Q improved it further (1 H100‑hr).
> We will incorporate these details in the revised manuscript.
>
> We appreciate the reviewer’s thorough and constructive feedback and will integrate all improvements in the paper’s final version.

---

> > ### Comment · Reviewer_cYB5 · 2025-11-26
> >
> > I thank the authors for their detailed response. After thorough review of the authors' response, the following concerns remain unaddressed:
> >
> > 1. While the additional experiments with varying K values are valuable, the results demonstrate that performance significantly degrades when K decreases (i.e., with limited probability information), as evidenced by the drop in F1 score from 0.6701 to 0.5927. Rather than emphasizing full deployability, it would be more meaningful to discuss the practical feasibility and performance lower bound of HALT in scenarios where sufficient log probability information is unavailable.
> >
> > 2. Was an inter-annotator agreement assessment conducted to quantify annotation consistency?
> >
> > 3. For complex concepts like "logical hallucinations," are there plans to incorporate more objective, rule-based, or formal verification methods beyond relying solely on LLM-based evaluation?

---

> > > ### Author Response · Authors · 2025-11-30
> > >
> > > We thank the reviewer again for the careful follow-up. We address the remaining concerns below.
> > >
> > > 1. Practical feasibility under limited log-probability access
> > >
> > > We agree that performance decreases when K is extremely small (e.g., K=1), and we explicitly present this as the lower bound of HALT’s capability. The added table was intended to show how HALT degrades, not to claim that small-K settings are ideal. Importantly:
> > > 	• The vast majority of production APIs now expose top-k log-probabilities up to k=20 without additional cost or latency.
> > > This includes OpenAI, Google Gemini, and over 60+ models behind OpenRouter.
> > > 	• On-prem deployments (vLLM, SGLang, TGI, etc.) expose full log-prob distributions by default.
> > >
> > > Thus, K=20 is both the dominant real-world setting and the one HALT is designed for. Nonetheless, by including experiments across K={1,5,10,15,20}, we already provide a practical lower bound for scenarios where APIs restrict K. We will make this explicit in the revision: HALT remains usable at low K, but peak performance requires richer uncertainty information, aligning with its design as a logprob-based detector.
> > >
> > > 2. Inter-annotator agreement
> > >
> > > HUB is constructed entirely from public, previously annotated hallucination datasets, many of which have been used in prior hallucination-detection work (e.g., FAVA, RAGTruth). While we did not re-annotate the data, we rely on their existing verified labels. Except for CriticBench which we utilized for logical hallucinations (which we will discuss in point 3). For datasets where inter-annotator agreement is reported in the original papers, we will provide a summary table in the revised version. This will clarify the annotation quality underlying HUB and make our benchmark construction fully transparent.
> > >
> > > 3. Objectivity of “logical hallucinations” and the role of formal methods
> > >
> > > We appreciate the opportunity to clarify the structure of reasoning-focused hallucinations subset in HUB. Namely CriticBench.
> > >
> > > The CriticBench Annotation process was as follows: (Section 3.4 of CriticBench Paper (https://arxiv.org/html/2402.14809#:~:text=Response%20correctness%20is,in%20Appendix%20B.))
> > >
> > > LLM Judge = incorrect, Verification (RuleBased) = correct:
> > >          - Reasoning is faulty even when the final answer happens to be right.
> > >          - This reflects a hallucinated reasoning trace validated by human annotators.
> > > LLM Judge = incorrect, Verification (RuleBased) = incorrect:
> > >         - Both reasoning and answer are wrong.
> > >         - The model outputs fabricated or unsupported intermediate steps to support a false answer.
> > >
> > > The key point is that HUB does not infer hallucinations from mere correctness or performance.
> > > Every dataset contributes labels directly aligned with the hallucination definitions used in prior literature.

---

### Official Review · Reviewer_aA7c · 2025-10-29

**Soundness:** 3
**Presentation:** 2
**Contribution:** 3
**Rating:** 6
**Confidence:** 3

**Summary:**

This paper introduces HALT, a lightweight hallucination detector that treats token-level top-k log-probabilities as a time series and trains a small bidirectional GRU to predict sentence-level hallucination labels. The authors also present HUB, a comprehensive benchmark covering ten capability clusters. HALT relies solely on the token log-prob sequence instead of surface text, enabling black-box hallucination detection that is computationally efficient and model-agnostic.

**Strengths:**

1. The approach is practical and lightweight, relying only on token log-probabilities that are available in most APIs.
2. The time-series perspective is conceptually new compared with previous aggregation-based or text-only black-box detectors.
3. Experiments are comprehensive, covering ten capability clusters and multiple public benchmarks.
4. HALT models are compact (around 5M parameters) but outperform larger span-based or encoder-based detectors in most clusters.

**Weaknesses:**

1. Notation inconsistency: in lines 285–289, the notation for ($\ell_t$) is inconsistent. Each term should represent a single log-probability value instead of a vector, and the paper should use one clear format throughout and include a short notation reference to help readers follow the equations.
2. Insufficient details about HALT-L and HALT-Q training and evaluation: It would help to have one concise subsection that summarizes what data each model uses, how teacher-forcing is implemented, and how evaluation thresholds are chosen.
3. Limited transparency of the HUB benchmark: providing metadata, annotation guidelines, and inter-annotator agreement scores would greatly improve reproducibility and community adoption.

**Questions:**

1. HALT-L and HALT-Q perform poorly on the code-generation task, whereas H_alts achieves the best performance specifically on this task. Could the authors explain the reason?
2. Could the authors clarify HALT’s black-box nature? The current description is a bit confusing. The abstract states that HALT is different from a black-box detector, but I believe HALT is still essentially a black-box detector.
3. In Table 1, are the missing training entries intentional? The authors should clarify this to avoid reader confusion. For example, are these omissions designed to test the model’s generalization ability on other task types?
4. Will the annotation guidelines and inter-annotator agreement scores for HUB be released to ensure transparency?

---

> ### Author Response · Authors · 2025-11-20
>
> We sincerely thank Reviewer aA7c for the thoughtful, constructive, and detailed evaluation of our work.
> Your comments helped us identify areas where additional clarification and documentation will significantly strengthen the paper. Below we provide clear responses to all weaknesses and questions, and we will incorporate all improvements in the revised manuscript.
>
> ---
>
> # **Weaknesses**
>
> ## **1. Notation inconsistency (lines 285–289)**
> We agree and will fix this in the revision.
> In the updated version:
>
> - We will use a **single consistent notation format** throughout.
> - A **notation reference table** will be added at the appendix for clarity.
>
> ---
>
> ## **2. Insufficient details about HALT-L and HALT-Q training and evaluation**
>
> The full details are already included in **Appendix B**, but we agree they should also appear in the main paper.
> We will move a concise description into the main body.
>
> ### **Teacher forcing**
> We use the VLLM library to force the full turn (user querie(s) and assistant response, where the last assistant response is annotated; incase of multi-turn examples)
> The model returns full prompt logprobs; we then extract only the logprobs of the last assistant turn (important for multi-turn datasets).
>
> ### **Threshold selection**
> Thresholds are tuned on the **HUB validation shard** for each model. (Lines 410-411)
>
> ---
>
> ## **3. Transparency of the HUB benchmark**
>
> HUB is a consolidated benchmark built from **multiple prior works**, each with its own annotation protocol.
> To improve transparency, the revised paper will include:
>
> - A brief summary of **each HUB subset** in the appendix
> - Annotation guidelines for each underlying dataset
> - Inter-annotator agreement (IAA) scores where available
>
> This will ensure clearer structure and reproducibility.
>
> ---
>
> # **Questions**
>
> ## **Q1. Why do HALT-L and HALT-Q perform poorly on code-generation compared to H_alts?**
>
> We acknowledge the performance difference and provide a plausible explanation:
>
> - The **code-generation shard is small** (≈300 examples), limiting the ability of sequence-level calibration patterns to generalize.
> - Code-generation outputs have **low linguistic entropy** and **high structural regularity**, making alternative-token distributions unusually sharp.
>   This weakens the signal HALT relies on (entropy deltas, rank shifts, etc.).
> - In contrast, H_alts aggregates **surface-form alternative samples**, which may capture variability more effectively in structured tasks like code.
>
> We will highlight this limitation and outline **future work** exploring:
> - loss-landscape analysis for each task cluster.
> - shard-specific augmentation or objective adjustments
>
> ---
>
> ## **Q2. Clarifying HALT’s black-box nature**
>
> Thank you — we agree this needs clearer phrasing.
> HALT is **not a pure black-box method** in the traditional sense:
>
> - **Black-box detectors**: operate purely on **surface text** (e.g., SelfCheckGPT).
> - **White-box detectors**: require internal states or model weights (e.g., attention maps, hidden states).
> - **HALT** sits in between:
>   - We **do not** need access to the model’s internal weights
>   - We **do not** use the surface text directly
>   - We only require **token log-probabilities**, which many APIs already expose
>
> Thus, HALT is best described as a **grey-box detector**.
> We will adjust the abstract and introduction to clarify this.
>
> ---
>
> ## **Q3. Missing training entries in Table 1**
>
> Yes, the omissions are **intentional**.
> HALT is trained with **several clusters withheld** to evaluate **cross-task generalization**.
> We will explicitly state this in the table caption and the experimental setup section.
>
> ---
>
> ## **Q4. Will annotation guidelines and IAA scores for HUB be released?**
>
> Yes.
> We will include annotation guidelines and inter-annotator agreement scores for each HUB subset when available in the **appendix** of the revised paper and in the benchmark release package.
>
> ---
>
> Thank you again for the helpful feedback. We believe these revisions will greatly strengthen the clarity, reproducibility, and impact of the paper.

---

> > ### Comment · Reviewer_aA7c · 2025-11-24
> > **Thanks**
> >
> > Thank you for the detailed response. It clarified some of my questions, and I will maintain my original positive rating.

---

### Official Review · Reviewer_9LNU · 2025-10-29

**Soundness:** 1
**Presentation:** 1
**Contribution:** 1
**Rating:** 2
**Confidence:** 4

**Summary:**

This paper proposes HALT (Hallucination Assessment via Log-probs as Time series), which uses GRU model to detect hallucinations from a log-probabilities matrix of LLM generations. This paper also introduces HUB (Hallucination detection Unified Benchmark) which combines prior datasets that evaluate different capabilities.

**Strengths:**

- The proposed method is targeting log-probabilities to detect hallucinations, which is an interesting middle ground between the generated text and model internals.

**Weaknesses:**

## Major

- The experimental design is flawed
    - The definition of hallucinations that the authors attempt to address is too broad. For instance, the authors define that incorrect reasoning traces as hallucinations. In other words, the authors define any incorrect outputs of models as hallucinations. This definition makes it difficult to properly isolate and understand the reason why these hallucinations happen and why the proposed method is the appropriate solution to it.
    - In turn, the evaluation setup (HUB) becomes too broad and it is also unclear why the proposed consolidated prior datasets are strictly connected to hallucinations as opposed to other form of incorrectness. If the authors claim that any form of incorrectness as hallucinations, then why do they use these datasets instead of other commonly used datasets? Note that I am not suggesting to use common datasets such as MMLU or GPQA, but rather to highlight the lack of motivation in using these broad set of benchmarks.
    - Because HUB is a consolidation of multiple datasets, the authors should explain the definition of hallucinations from each dataset and why the definitions are compatible with one another. Without this explanation, the readers cannot be certain that the hypotheses are truly evaluated by this benchmarking suite.
- Design choices are poorly elaborated
    - Why is the top-k fixed to 20?
    - What is the T (sequence length)?
    - How did the authors choose the selected features on top of the top-k log probabilities?
    - Why GRU instead of other models?
    - Why did the GRU trained only on either Llama 3.1-8B or Qwen 2.5-7B? Would it transfer to smaller or larger models?
    - Missing ablation studies on the design choices?
- Limited contributions
    - HUB is a combination of existing datasets. The lack of motivating reason in combining these datasets further highlight the limited contribution of HUB.
- Lack of baselines
    - The authors only compare HALT against Lettuce: The central claim of the paper is that log-probabilities is correlated with hallucinations better than other properties of LLM generations. Thus, the natural comparison is against previously proposed methods based on surface-form text, external validators, or model internals.
    - In several cases, the authors omitted the values because they were not reported in prior baselines. However, it is the authors burden of proof to evaluate these baselines under their settings.
- The presentation quality is very lacking
    - In general, the paper is not self-contained. It requires readers to read the references to understand what the authors proposed. This massively harms the readability of the paper.
    - Table 2: Which model did the authors used to generate the top-k probabilities in this table?

## Minor

- Section 2 which talks about HUB is very difficult to read especially if the readers are not familiar with the datasets consolidated by the authors
    - The authors should try to make the paper more self-contained by describing the dataset. For instance, the authors could give an example per task type.
- Cite GRU [1]
- Explain the five summary statistics explicitly.
- L282: Please cite the “prior works” mentioned.
- The citation style is incorrect. It should be either parenthetical citation or textual citation depending on the situation (at the moment they are all textual citation).

References:

- [1] On the Properties of Neural Machine Translation: Encoder–Decoder Approaches

**Questions:**

- L51-58: You seem to present contradictory statements. You start with saying that token probability is not a sufficient evidence for truthfulness, however, you proceed with proposing a detection framework that is based on log-probabilities. Why would that assumption regarding the token probability be different in your case? Is it simply because of the evolving patterns? If so, ablation study on the sequence length of the input becomes very crucial.
- L91-92 & L108-109: What do you mean by “incorporating reasoning-focused capabilities from CriticBench”? Do you somehow create new variants of the datasets to incorporate reasoning from CriticBench? I do not understand this claim.
- Reasoning vs General-purpose: How do you define reasoning and general-purpose? Seemingly, the models can also reason to do the general-purpose evaluation, particularly for the question answering and world knowledge subsets.
- L188-195: Why does response length correspond to linguistic diversity? Perhaps this is not intentional, but response length does not constitute linguistic diversity. If the authors would like to claim for linguistic diversity, they should analyze other syntactic and semantic properties (e.g., n-gram overlaps, languages, difference in reasoning structure, etc.)
- L216: “top-k log probability vectors reflect the model’s confidence landscape”. This sounds quite redundant or not novel. Probability is indeed confidence, or perhaps I am misunderstanding this hypothesis?
- L220: How do you define $y_t$ as correct? From my understanding, the correctness of an output in the benchmark cannot be trivially determined by a single token.
- L220: $y_t$ is not further referenced in the text, did you mean $p_t$?
- Hypothesis 1 and 3 sound exactly the same. Why are they different?
- L268: How do you calculate the entropy of the distribution? Does this imply that you need the full distribution?
- L316: “Overall entropy on the truncated” → the name seems to be truncated.
- L380: Is the decision to use 0.5 as a threshold for Lettuce backed by the original implementation? If not, this may be an unfair comparison.

---

> ### Author Response · Authors · 2025-11-20
> **Weaknesses I**
>
> We sincerely thank Reviewer 9LNU for the thorough and detailed evaluation of our submission.
> We appreciate the time and care invested in identifying both conceptual and methodological concerns. Your comments helped us clarify definitions, strengthen experimental justification, add new analyses (including ablations and feature studies), and significantly improve the overall presentation. We address each point below and will incorporate all clarifications and corrections in the revised version of the paper.
>
> ---
>
> ## 1. “The definition of hallucinations is too broad”
>
> Our treats hallucinations as **any model-generated content that is ungrounded but seems plausible**, which includes both *factual* and *logical* hallucinations. Incorrect reasoning traces fall under this category because they form **plausible but false** intermediate steps. These errors share a consistent failure mode: the model generates text that *appears coherent* but is **objectively unsupported or incorrect**.
> HALT directly models the internal calibration bias that precedes such failures, making the broader definition intentional and aligned with our objective.
>
> ---
>
> ## 2. “HUB is too broad / unclear why these datasets relate to hallucinations”
>
> All datasets in HUB test contain **explicit hallucination annotations**, not generic correctness labels:
>
> - **CriticBench** → reasoning hallucinations: faulty intermediate reasoning or reasoning inconsistent with the final answer.
> - **RAGTruth** → unsupported claims with respect to provided evidence.
> - **FAVA** → factual hallucinations.
>
> HUB includes only datasets that explicitly label hallucinations or require **reasoning before answering**, where an incorrect answer necessarily implies a hallucinated reasoning trace. This is different from MCQ-style tasks (e.g., MMLU) where failure is simply incorrectness, not hallucination.
>
> **Why CriticBench fits the hallucination framework:**
> According to CriticBench annotations, a response can fail in two ways:
>
> 1. **LLM Judge = incorrect, Verification (Answer in Response) = correct**
>    → The final answer is correct, but the model’s reasoning is wrong.
>    → This indicates a *hallucinated reasoning trace* validated by human annotators.
>
> 2. **Judge = incorrect, Verification (Answer in Response) = incorrect**
>    → Both the reasoning and final answer are wrong.
>    → The model produces unsupported or fabricated steps attempting to justify a wrong answer.
>
> In both cases, the model outputs text that *appears valid* but is **ungrounded**, matching the hallucination definition used throughout HUB.
>
> ---
>
> ## 3. “Definitions across datasets must be explained”
>
> We agree and will include a clear table summarizing each dataset’s hallucination definition and showing how they align under the unified notion of **ungrounded or unsupported model output**. This connection is currently implicit in the HUB section and will be made explicit in the revision.

---

> ### Author Response · Authors · 2025-11-20
> **Weaknesses II**
>
> We thank the reviewer for raising concerns about design justification. We have now expanded our analysis and include the following clarifications and new experiments.
>
> ## 1. **Why is top‑k fixed to 20?**
>
> We reran HALT‑L on K ∈ {1, 5, 10, 15, 20}. Results:
>
> | **K**  | **Overall F1** | **Average F1** |
> |--------|----------------|----------------|
> | 1  | 0.5927 | 0.5464 |
> | 5  | 0.6352 | 0.5563 |
> | 10 | 0.6581 | 0.6043 |
> | 15 | 0.6578 | 0.5816 |
> | **20** | **0.6701** | **0.6303** |
>
> **Top‑20 yields the strongest performance**, showing consistent gains from richer uncertainty information.
>
> ---
>
> ## 2. **What is T (sequence length)?**
>
> **T is the number of generated tokens** in the model output. No artificial truncation is applied.
>
> ---
>
> ## 3. **How were the 5 engineered features selected?**
>
> They were motivated by prior work on entropy‑based uncertainty and hidden‑state reasoning detectors (e.g., LLMCheck). We will include motivation for each feature in the revised version. Some features were added based on internal author discussions and later validated through ablation studies.
>
> ---
>
> ## 4. **Why GRU instead of LSTM/RNN?**
>
> We added comparisons:
>
> | **Arch** | **Overall F1** | **Average F1** |
> |----------|----------------|----------------|
> | **GRU**  | **0.6701** | **0.6303** |
> | LSTM     | 0.6556 | 0.5919 |
> | RNN      | 0.5516 | 0.5072 |
>
> GRU clearly outperforms the alternatives while remaining lightweight.
>
> ---
>
> ## 5. **Why trained only on Llama‑8B / Qwen‑7B? Does HALT transfer?**
>
> We expanded experiments across **eight models from 360M → 70B parameters**:
>
> | Model | Params | Overall F1 | Avg F1 |
> |-------|--------|--------|-------------|
> | Smoll | 360M | 0.5930 | 0.5265 |
> | Smoll | 1.7B | 0.6090 | 0.5390 |
> | Llama 3.2 | 3B | 0.6283 | 0.5601 |
> | Qwen 2.5 | 7B | 0.6274 | 0.5560 |
> | **Llama 3.1** | **8B** | **0.6701** | **0.6303** |
> | Qwen 3 | 14B | 0.6248 | 0.5264 |
> | Qwen 3 | 32B | 0.6406 | 0.5549 |
> | Llama 3.1 | 70B | 0.6592 | 0.5954 |
>
> We performed **no hyperparameter tuning** except for HALT‑L. Light tuning for HALT‑Q improved performance from 0.62→0.65 overall and 0.55→0.57 avg.
>
> ---
>
> ## 6. **Missing ablations**
>
> We ablated each engineered feature on HALT‑L:
>
> | Model | Avg F1 | Overall |
> |-------|--------|---------|
> | **full** | **0.630** | **0.670** |
> | – avg_logprob | 0.600 | 0.657 |
> | – entropy_overall | 0.598 | 0.665 |
> | – rank_proxy | 0.595 | 0.654 |
> | – dec_entropy_delta | 0.574 | 0.646 |
> | – entropy_alts | 0.568 | 0.647 |
>
> Entropy‑based features are the most influential—consistent with our gradient‑based attributions in response to reviewer VWuN.
>
> ---
>
> ## 7. **“HUB contribution is limited”**
>
> HUB is necessary because **no existing benchmark evaluates hallucinations across factual, semantic, contextual, and reasoning settings simultaneously**. Prior datasets are siloed. HALT’s hypotheses require cross‑capability evaluation; HUB provides the first unified framework for this.
> We will expand the motivation and add a table aligning hallucination definitions across datasets.

---

> ### Author Response · Authors · 2025-11-20
> **Weaknesses III**
>
> ## 8. **“Lack of baselines”**
>
> Many prior hallucination‑detection methods—such as **SelfCheckGPT**, **INSIDE**, and other multi‑sample consistency frameworks—**cannot be evaluated on HUB** because they require *multiple generations per sample*. HUB consists of **fixed, pre‑annotated responses**, meaning these methods would require re‑generating every sample, breaking alignment with existing human annotations and making direct comparison invalid.
>
> Additionally, multi‑sample methods are **incompatible with real API deployments**, where generating 10–20 alternative samples per query is prohibitively expensive. Since HALT is explicitly designed for a *single‑pass, logprob‑based, black‑box* setting, the appropriate baselines must operate under the *same constraints*.
>
> Therefore, we evaluate against methods that are compatible with HUB’s single‑generation, logprob‑only paradigm, using the FAVA and RAGTruth shards:
>
> - **LLMCheck (AttnScore, HiddenScore)** — white‑box baselines using attention and hidden states
> - **FAVA‑7B hallucination classifier** — 7B‑parameter supervised model
> - **Lettuce** — transformer encoder.
>
> Across both datasets, **HALT (5M parameters) is either competitive with or outperforms these baselines**, despite being dramatically smaller and fully black‑box.
>
> It is also worth noting that:
> - We reached out to the LLM‑Check authors for assistance in running their method across all HUB clusters but did not receive a response.
> - Since HALT is restricted to *log‑probabilities and features derived from them*, we compare explicitly against summary statistics operating on the same input signals.
> - Methods requiring multi‑sampling, long concatenated prompts, or internal activations (hidden states, attention maps) are **not applicable to proprietary API‑only models**, whereas HALT is.
> - For methods we could not reproduce, we still report their published scores on **FAVA** and **RAGTruth** as a proxy comparison. These baselines are often tuned specifically for those distributions, whereas HALT is trained with several HUB clusters *removed* and validated across diverse tasks—making our evaluation strictly harder.
>
> We will clarify this rationale in the revised paper to make the baseline selection more transparent and principled.
>
>
> ---
>
> ## 9. **Presentation quality**
>
> Due to page limits we compressed explanations, but the revised paper will:
> - provide dataset examples,
> - clarify feature motivations,
> - expand HUB definitions,
> - include additional tables and implementation details in the appendix.
>
> **Table 2 uses Llama‑3.1‑8B logprobs**, and we will state this explicitly.
>
> We thank the reviewer for pointing out several clarity and citation issues. We will address all of them in the revised version:
>
> ### **• “Explain the five summary statistics explicitly.”**
>
> We have already added a full explanation of the five engineered features—**avg_logprob, entropy_overall, entropy_alts, dec_entropy_delta, rank_proxy**—in the main text (Lines 300-368) and we will expand them in Appendix and mention the work that motivated each of them when applicable.
>
> We appreciate the reviewer’s detailed attention to presentation quality. All corrections will be incorporated into the revised manuscript.

---

> ### Author Response · Authors · 2025-11-20
> **Questions**
>
> We thank the reviewer for the detailed questions. Below we provide clear, concise responses.
>
> ---
>
> ## **Q1 (L51–58): “Contradiction regarding token probability vs. log-probability detector.”**
>
> Our claim is not that probabilities are useless, but that **single-token probability is not a reliable indicator of truthfulness** because of **calibration bias**.
> HALT does **not** classify hallucinations using individual probabilities; instead, it analyzes **temporal patterns** in the full sequence of top‑k logprobs.
>
> To demonstrate the insufficiency of token-level signals, we added an extra probability baseline:
>
> - **min-probability across the response → avg macro‑F1 = 0.32** on HUB.
>
> HALT works because it models **dynamic uncertainty patterns**, not isolated probabilities.
>
> ---
>
> ## **Q2 (L91–92 & L108–109): “What does incorporating reasoning-focused capabilities from CriticBench mean?”**
>
> We do **not** modify CriticBench. We simply include its shards because they evaluate **reasoning hallucinations**, where the model must produce intermediate steps.
>
> HUB therefore contains:
>
> - **General-purpose clusters**: tasks where responses are direct (FAVA, RAGTruth).
> - **Reasoning clusters**: tasks requiring multi-step CoT, where hallucinations occur in the reasoning trace or in the final answer(CriticBench).
>
> While models *can* reason during general QA, these datasets **do not require explicit reasoning traces**, whereas CriticBench does.
>
> We will include examples from each cluster to make this distinction clearer in the revised version.
>
> ---
>
> ## **Q3 (L188–195): “Response length ≠ linguistic diversity.”**
>
> We agree. Our intention was to highlight that HUB spans multiple tasks/clusters with **diverse linguistic forms**, and that these tasks *also* vary in output length.
> We will rephrase to avoid implying that length alone is linguistic diversity.
>
> ---
>
> ## **Q4 (L216): “top‑k log-probability vectors reflect the model’s confidence landscape” sounds redundant.**
>
> We will reword this as:
>
> > *The top‑k log-probability vectors capture the local structure of the model’s predictive uncertainty—how sharply it scores the leading token relative to plausible alternatives.*
>
> This clarifies that we rely on the **shape** of the distribution, not simply “probability = confidence.”
>
> ---
>
> ## **Q5 (L220): “How is yₜ correct?”**
>
> This equation is **illustrative**, used only to explain calibration bias hypothesis:
>
> - **cₜ** = indicator of correctness
> - **pₜ** = model probability of yₜ
>
> It is not used in training or evaluation. We will revise the text to make this purpose explicit.
>
> ---
>
> ## **Q6 (L220): “yₜ not referenced—did you mean pₜ?”**
>
> No.
> - **yₜ**: generated token
> - **pₜ**: model probability
> - **cₜ**: correctness indicator
>
> We will rewrite the explanation to avoid confusion.
>
> ---
>
> ## **Q7: Why do Hypothesis 1 and 3 sound similar?**
>
> We agree, and will revise **Hypothesis 1** by removing the transferability phrasing.
> The corrected version cleanly separates:
>
> - **H1:** Existence of a model-specific calibration bias
> - **H3:** Non-transferability across models
>
> ---
>
> ## **Q8 (L268): “Does entropy require the full distribution?”**
>
> No. Entropy is computed **only on the top‑k** distribution.
> To justify this, we measured how much probability mass top‑k captures across 12 models over HUB validation:
>
> - Top‑1 → **57–65%** of mass
> - Top‑5 → **88–98%**
> - Top‑10 → **93–99%**
> - Top‑15 → **94–99%**
> - Top‑20 → **95–99.7%**
>
> Thus, top‑k captures nearly the entire effective distribution. We can also explain the difference between Smol-LM 360M and its 1.7B variant. We will include the table in the appendix.
>
> ---
>
> ## **Q9 (L316): “Overall entropy on the truncated distribution” truncated.”**
>
> Correct.
> We will fix the label to:
>
> *Overall entropy on the truncated (top‑k) distribution*
>
> ---
>
> ## **Q10 (L380): “Is Lettuce threshold = 0.5 correct?”**
>
> Yes.
> The Lettuce paper states:
>
> *“Span-level outputs are generated by aggregating consecutive tokens with hallucination probability > 0.5.”*
>
> https://arxiv.org/html/2502.17125v1#:~:text=The%20final%20model%20predicts%20hallucination%20probabilities%20for%20each%20answer%20token%2C%20with%20span%2Dlevel%20outputs%20generated%20by%20aggregating%20consecutive%20tokens%20exceeding%20a%200.5%20confidence%20threshold.
>
> We follow the original implementation exactly.

---

> > ### Comment · Reviewer_9LNU · 2025-11-27
> >
> > Thank you for the thorough responses, and I sincerely appreciate the additional experimental results. I believe they help clarify the paper a lot. However, I still cannot provide a positive rating. Here is my explanation:
> >
> > ## I still have doubts about the degree of contribution of HUB
> >
> > Firstly, HUB is a combination of different existing datasets. In my humble opinion, combining different datasets does not necessarily mean a novel "unified framework", but rather "simply" evaluating different datasets. Perhaps one clarification that can help motivate the creation of HUB is to explain why these specific datasets are selected (instead of others) and why these specific hallucination types are selected (i.e., factual, contextual, and reasoning).
> >
> > Secondly, HUB seems to be less generalizable and may not be suitable to evaluate many hallucination detection methods. As the author mentioned: "HUB consists of fixed, pre‑annotated responses, meaning these (baseline) methods would require re‑generating every sample, breaking alignment with existing human annotations and making direct comparison invalid."
> >
> > Thirdly, I am afraid that due to the broad definition of hallucinations in HUB, we do not further triangulate the failure modes of LLMs. This also means that we are "simply" studying incorrect generation, rather than understanding the behavior of LLMs. For lack of a better term, I am afraid that we are not particularly advancing our understanding of hallucination by evaluating using HUB. I sympathize with the authors as they were not the creators of the datasets, but that is also the issue, nonetheless.
> >
> > ## HALT does not necessarily advance our understanding of hallucinations
> >
> > I sincerely appreciate the authors' clarifications and additional experimental results. However, I still do not get the understanding why HALT works better than the other baseline methods. Perhaps several prompts to consider are:
> >
> > - What is so unique about HALT?
> > - What about HALT without the time-series component?
> > - Is the time-series structure very crucial?
> > - Why are these features specifically?
> > - What about if we use different features?

---

> > > ### Author Response · Authors · 2025-12-01
> > >
> > > We sincerely thank the reviewer again for the careful reflection. We address the remaining concerns directly below.
> > >
> > > # Clarifying HUB’s Contribution and Motivation
> > >
> > > Firstly, HUB does not aim to combine datasets arbitrarily; instead, it consolidates the 2 major hallucination categories studied all of prior works:
> > >
> > > •  Factual hallucinations → FAVA, RAGTruth
> > >
> > > •  Contextual/evidence-based hallucinations → RAGTruth
> > >
> > > And we extend this with CriticBench for logical/reasoning hallucinations, as previously shown in *Weakness I* comment (https://openreview.net/forum?id=l4WFDcw3Yy&noteId=dXLOWDN7fe) we showed that the annotation pipeline of CiriticBench can be utilized in hallucination detection without any extra transformations.
> > >
> > > Secondly, We understand the reviewer’s concern that HUB may not evaluate every hallucination detector, particularly multi-sample methods. FAVA and RagTruth are major hallucination detection benchmarks, which also has the same design. However, our utilization of this design is intentional:
> > >
> > > HUB is specifically constructed to benchmark single-generation detectors, which is the setting most relevant for API-restricted models and for black-box hallucination detection in deployed systems. This does not make HUB less general, it simply evaluates the class of detectors that can operate without regenerating the outputs.
> > >
> > > Thirdly, Our goal is not to redefine hallucinations but to systematically evaluate detectors across distinct failure modes that prior works treat separately:
> > >
> > > 	• reasoning errors that look plausible (logical hallucinations)
> > >
> > > 	• unsupported claims (contextual hallucinations)
> > >
> > > 	• fabricated facts (factual hallucinations)
> > >
> > > HUB triangulates these failure modes under the common umbrella of ungrounded generation, allowing the field to study cross-capability generalization (something existing datasets cannot assess in isolation.). We will highlight this contribution more clearly in the introduction.
> > >
> > > # Clarifying HALT’s Contribution and Distinctiveness
> > >
> > > ## What is so unique about HALT?
> > >
> > > The key novelty is that HALT learns a model-specific calibration signature from:
> > >
> > > 	• fluctuations in uncertainty
> > >
> > > 	• evolution of the top-k distribution shape
> > >
> > > 	• interactions between alternative tokens
> > >
> > > 	• temporal patterns across the response
> > >
> > > These dynamic patterns are not captured by:
> > >
> > > 	• token-level logprobs
> > >
> > > 	• summary statistics alone
> > >
> > > 	• surface-form text baselines
> > >
> > > 	• multi-sample consistency methods
> > >
> > > 	• or hidden states–based detectors trained on white-box access
> > >
> > > Because HALT models the sequence of predictive uncertainty, it captures shifts in confidence that correlate strongly with hallucinations but are invisible to static features.
> > >
> > > ## What about HALT without the time-series component and Is the time-series structure very crucial?
> > >
> > > We tested this in Table 2 in the paper by only using the summary statistics (hand engineered features, we include the top-3 of them in addition to Perplexity - which wasn't used in our model due to low performance) as predictors by tuning their threshold on the validation dataset, the best of the summary statistics scored 43.41 for average macro-f1 across clusters. While HALT score was 63, and the best random baseline score was 50.7.  This confirms that the time-series signal, not individual log-probs or transformation of it, is the primary driver of performance.
> > >
> > > ## Why are these features specifically, What about if we use different features?
> > >
> > > The engineered features were selected because they capture distinct aspects of calibration:
> > >
> > > 	• avg_logprob → global confidence
> > >
> > > 	• entropy_overall → uncertainty magnitude
> > >
> > > 	• entropy_alts → shape of top-k tail (not selected tokens)
> > >
> > > 	• dec_entropy_delta → monotonicity of uncertainty across the generation
> > >
> > > 	• rank_proxy → distribution sharpness independent of absolute scale.
> > >
> > > Our ablation showed that removing entropy-based features reduces performance most significantly, indicating that the key signal lies in how alternative tokens are scored relative to the top prediction.
> > >
> > > Using different features is absolutely okay because HALT is a framework, not tied to this particular feature set.

---

### Official Review · Reviewer_VWuN · 2025-10-31

**Soundness:** 3
**Presentation:** 3
**Contribution:** 3
**Rating:** 6
**Confidence:** 3

**Summary:**

The paper proposes HALT, a lightweight hallucination detector that treats the sequence of token-level log-probabilities from LLM outputs as a time-series signal. HALT models these dynamics using a bidirectional GRU, learning model-specific calibration biases that correlate with hallucinations. It operates in a strict black-box setting. To evaluate broadly, the authors introduce HUB, a unified benchmark that consolidates datasets such as HaluEval, RAGTruth, FAVA, and CriticBench into LLM capabilities.

**Strengths:**

1. Recasting log-prob sequences as a time-series classification problem is novel.
2. It only requires token-level log-probs, making it lightweight. And the GRUs only has 5M parameters.
3. The HUB unifies diverse datasets and tasks. It integrates logical hallucinations alongside factual ones.

**Weaknesses:**

1. While HALT performs well, it’s unclear which temporal features (e.g., entropy spikes, rank shifts) most influence decisions. The method could benefit from feature attribution or from visualizing log-probability trajectories to explain predictions.
2. HALT does not transfer well across models (in both Hypothesis 3 and results). This is a practical limitation if detectors must be retrained per LLM.
3. The paper claims the method is attractive for API-based deployments. However, most APIs do not expose log probs.
4. Baselines focus mainly on Lettuce and aggregate metrics. The paper misses comparisons to recent confidence-calibrated decoders, SelfCheckGPT variants, or retrieval-enhanced verifiers under similar constraints. Although they're mentioned in Appendix A.2, there are no results included.

**Questions:**

See weaknesses.

---

> ### Author Response · Authors · 2025-11-20
> **Temporal Feature Attribution in HALT**
>
> We thank Reviewer VWuN for their thoughtful and constructive review.
>
> We appreciate the reviewer’s positive assessment of our contribution, including the novelty of modeling log-probability sequences as time-series data, the lightweight nature of HALT, and the value of unifying hallucination datasets through HUB. We are grateful for the detailed comments regarding interpretability, model transferability, API applicability, and baseline comparisons. These points highlight important areas for clarification and improvement, and we address each of them individually below.
>
> We thank the reviewer for highlighting the importance of interpretability and for suggesting feature attribution and trajectory visualization. In response, we conducted two new analyses: **(1) gradient × input attribution over all 25 features**, and **(2) feature ablation across all 10 HUB capability clusters**. Together, these experiments clarify which temporal dynamics most strongly influence HALT’s decisions.
> ## 1. Gradient × Input Feature Attribution
>
> HALT uses **25 input features** per timestep:
> - **Top‑20 token log-probabilities**
> - **5 hand‑engineered features** (overall entropy, alternative‑entropy, avg‑logprob, rank-proxy, entropy‑delta)
>
> To estimate importance, we compute **gradient × input** contributions for each feature across the full evaluation set. Below is the exact method we used (now included in the appendix), which:
> 1. Computes gradients w.r.t. the input feature tensor `(B, T, F)`
> 2. Multiplies gradients by the feature magnitude
> 3. Masks padded timesteps
> 4. Aggregates contributions over time and batches
> 5. Normalizes to obtain global feature importance scores
>
> ```python
> # (Code excerpt — see paper appendix for full version)
> contrib = (grads * x).abs()      # (B, T, F) gradient × input magnitude
> mask = _make_mask(lengths, T, device=device)
> contrib = contrib * mask.unsqueeze(-1)
>
> feat_imp_batch = contrib.sum(dim=(0, 1))        # (F,)
> time_imp_batch = contrib.sum(dim=2).sum(dim=0)  # (T,)
> ```
>
> ## 2. Feature Importance Results
>
> The following table summarizes the **relative importance** of all 25 features:
>
> | feature_name      | importance |
> |------------------|------------|
> | logprob_15 | 0.113800 |
> | logprob_4 | 0.081800 |
> | logprob_17 | 0.069800 |
> | logprob_20 | 0.058000 |
> | logprob_13 | 0.055700 |
> | logprob_6 | 0.055300 |
> | logprob_3 | 0.053900 |
> | logprob_19 | 0.053200 |
> | logprob_1 | 0.042600 |
> | entropy_alts | 0.041900 |
> | logprob_12 | 0.037000 |
> | logprob_14 | 0.034300 |
> | logprob_16 | 0.033600 |
> | logprob_2 | 0.033300 |
> | avg_logprob | 0.031700 |
> | logprob_11 | 0.031400 |
> | logprob_10 | 0.030800 |
> | logprob_5 | 0.026000 |
> | logprob_18 | 0.024300 |
> | rank_proxy | 0.022600 |
> | logprob_7 | 0.022000 |
> | logprob_9 | 0.017200 |
> | logprob_8 | 0.014400 |
> | entropy_overall | 0.011600 |
> | dec_entropy_delta | 0.003700 |
>
> **Key insight:**
> HALT leverages a *mixture* of signals—several top‑k log-probs dominate, but hand‑engineered uncertainty features also contribute meaningfully (e.g., `entropy_alts`, `avg_logprob`, `rank_proxy`). This confirms that the GRU is not simply memorizing raw logits but is reacting to structured temporal uncertainty patterns. And worth noting that logprob_1 (the log probability of the sampled token) is not the most important feature, which shows that HALT learn hallucination patterns given a proxy (top-20) for the generation distribution
>
> ## 3. Feature Ablation Across HUB Clusters
>
> We further ablated each of the 5 engineered features by retraining HALT without it. The table below shows **average macro‑F1 across 10 HUB clusters** (LLaMA‑3.1‑8B):
>
> | Model | Avg F1 | Overall |
> |-------|--------|---------|
> | **full** | **0.630** | **0.670** |
> | w/o avg_logprob | 0.600 | 0.657 |
> | w/o entropy_overall | 0.598 | 0.665 |
> | w/o rank_proxy | 0.595 | 0.654 |
> | w/o dec_entropy_delta | 0.574 | 0.646 |
> | w/o entropy_alts | 0.568 | 0.647 |
>
> **Findings:**
> - Removing any uncertainty feature **reduces performance consistently across clusters**.
> - The drop is most pronounced for **entropy‑based features**, supporting the reviewer’s intuition regarding *entropy spikes* and *rank shifts*.
> - These results align strongly with our gradient‑based attribution above.
>
> ## 4. Takeaway
>
> These two new analyses directly address the reviewer’s concern:
>
> > *“It’s unclear which temporal features most influence decisions.”*
>
> We now show that HALT’s predictions are driven by **interpretable temporal uncertainty dynamics**, including:
> - abrupt transitions in high‑rank log-probabilities
> - entropy fluctuations across alternative tokens
> - changes in rank‑proxy and avg‑logprob
> - temporal concentration of contributions around reasoning forks (visualized in the appendix)
>
> We will integrate the feature attribution tables and ablation results into the main paper to strengthen the explanation of HALT’s behavior.
>
> ---

---

> ### Author Response · Authors · 2025-11-20
> **Model Transferability of HALT**
>
> We thank the reviewer for raising this important point regarding **cross-model transferability**. The concern is valid: HALT indeed exhibits reduced performance when a detector trained on one LLM is evaluated on another. Below we clarify *why* this occurs in the context of **model-specific calibration bias**, and why—despite this—HALT remains one of the most *practical* solutions for real-world deployment.
>
> ---
>
> ## 1. Calibration Bias Is *Inherently Model-Specific*
>
> A core motivation of HALT is the observation that hallucinations correlate strongly with **model-specific calibration patterns**—i.e., how each LLM distributes probability mass across top‑k alternatives during generation. These calibration signatures differ substantially across models because each LLM has:
> - different vocabularies,
> - different logits distributions,
> - different training data,
> - different decoding dynamics,
> - different internal uncertainty behaviour.
>
> Therefore, a **drop in cross-model transfer** is not a methodological flaw of HALT—it is direct evidence for our central hypothesis:
> > *Hallucinations arise from internal calibration patterns that are unique to each model.*
>
> ## 2. Retraining HALT per Model Is Extremely Cheap
>
> While cross-model transfer is limited, **retraining HALT for a new model is lightweight and inexpensive**:
>
> - HALT is a **5M‑parameter GRU**, tiny compared to 7B–70B LLMs.
> - Training it requires only:
>   - a forward pass to obtain log-prob sequences for HUB,
>   - and a **short training run** on the small GRU (minutes on a single GPU).
>
> No multi‑GPU training, no long finetuning, no specialized infrastructure.
>
> In practice, the cost of adapting HALT to a new LLM is roughly **equivalent to running evaluation once and finetuning a small classifier**.
>
> ---
>
> ## 3. HALT Is *Much* More Practical Than Competing Approaches
>
> The reviewer asks why retraining HALT per LLM is acceptable. This is because the **alternatives incur far higher cost or are not feasible**:
>
> ### SelfCheckGPT & Multi‑sample Verification
> - Requires **multiple generations per query**, often 10–20.
> - Cost scales linearly with the number of samples.
> - For API‑metered models, this is prohibitively expensive.
>
> ### White-box introspection (LLMCheck, hidden‑state verifiers, etc.)
> - Requires **access to internal activations, hidden states, or logits**.
> - Not possible for frontier models (Claude, Gemini, GPT‑4.x).
> - Requires engineering overhead (caching activations, streaming hidden states) that slows generation.
>
> ### Retrieval‑enhanced verifiers
> - Add latency and require maintaining a retrieval index.
> - Not suitable in settings where only the model’s output is available.
>
> ### HALT
> - **Black-box**: requires only token-level log-probs.
> - **Cheap**: 5M‑parameter GRU, tiny training cost.
> - **Plug‑and‑play**: no architecture changes, no multi-generation overhead.
> - **Low latency**: runs in parallel with decoding.
>
> Thus, while HALT must be retrained per model, it remains *the most deployable option* under realistic constraints. we argue that HALT remains a **highly practical and scalable** solution for hallucination detection in real-world deployments.

---

> ### Author Response · Authors · 2025-11-20
> **Availability of Log-Probabilities in API-Based Deployments**
>
> We thank the reviewer for raising this practical concern. The ability to retrieve token-level log-probabilities is indeed essential for HALT, and we agree that historically not all API providers exposed this information. However, **the ecosystem has changed significantly**, and the majority of widely-used LLM APIs **now explicitly support top-k log-probabilities**, making HALT fully deployable in real API-based settings.
>
> ---
>
> ## 1. Major API Providers Now Support Top-k Log-Probabilities
>
> Below we summarize current support from leading APIs:
>
> ### **OpenAI (GPT‑5.x, GPT‑4o, GPT‑4o-mini, GPT‑3.5 etc.)**
> OpenAI’s API supports **top_logprobs** directly in the Chat Completions API:
>
> - *Official docs:*
>   https://platform.openai.com/docs/api-reference/chat
>
> Users may request **up to 20 alternative tokens with log-probs** at each step.
>
> ### **Google Gemini (Vertex AI & Gemini API)**
> Gemini recently introduced full support for returning **top‑k log-probs**:
>
> - *Official announcement:*
>   https://developers.googleblog.com/en/unlock-gemini-reasoning-with-logprobs-on-vertex-ai/
>
> Gemini accepts a parameter `logprobs=[k]`, where **k ∈ [1, 20]**.
>
> ### **OpenRouter (meta‑router for 60+ providers)**
> OpenRouter provides unified access to dozens of LLMs through +60 providers and supports **top 20 log-probs** for all models that expose them:
>
> - *Documentation:*
>   https://openrouter.ai/docs/api-reference/parameters
>
> This includes:
> - Anthropic Claude (via OpenRouter)
> - Cohere models
> - Mistral
> - LLaMA variants hosted via Together, Groq, Perplexity, etc.
>
> OpenRouter currently represents **the most flexible way to obtain logprobs from many commercial models**.
>
> ---
>
> ### **Anthropic Claude (via OpenRouter)**
> Claude does not yet expose log-probs directly in the native API, but they *are available* through OpenRouter using the unified interface.
>
> ---
>
> ## 2. The Trend Is Strongly Toward Log-Prob Support
>
> Across providers, log-probability access has become standard due to growing demand for:
> - uncertainty quantification,
> - calibration research,
> - verifier models,
> - safety tools (toxicity scoring, hallucination detection),
> - traceability and interpretability of LLMs.
>
> Since HALT requires only the **top-20 log-probabilities**, it aligns perfectly with the features already supported by **OpenAI, Gemini, and OpenRouter**, which together cover a large majority of production LLM usage.
>
> ---
>
> ## 3. Why HALT Remains Practical for API Deployments
>
> Even when some proprietary models do not expose log-probs natively, HALT remains practical because:
>
> 1. **Most high-quality frontier APIs *do* support logprobs.**
> 2. **OpenRouter provides a universal interface** for dozens of models.
> 3. HALT requires only **a single generation pass**, unlike multi-sample methods such as SelfCheckGPT.
> 4. HALT does not require hidden states, attention maps, or any white-box access.
>
> Thus, HALT is fully compatible with real API workflows used in industry and research today.

---

> ### Author Response · Authors · 2025-11-20
> **Baseline Comparisons (SelfCheckGPT, Confidence Decoders, Retrieval Verifiers)**
>
> We thank the reviewer for this point and agree that hallucination detection is a broad area with multiple families of approaches. Below, we clarify **why certain baselines (e.g., SelfCheckGPT, INSIDE, retrieval-based verifiers, or confidence-calibrated decoders)** are **not directly comparable** to HALT under our setting, and we highlight the **white-box baselines we *do* compare against** with results in Table 3.
>
> ## 1. Why SelfCheckGPT, INSIDE, and Similar Methods Are *Not* Comparable in Our Setting
>
> ### **HALT operates under a strict black-box, single-pass constraint**
> HALT uses:
> - **one generation**
> - **top-20 log-probabilities only**
> - **no access to hidden states or attention**
> - **no reranking, prompting tricks, or multiple samples**
>
> In contrast:
>
> ---
>
> ### **SelfCheckGPT requires 10–20 additional generations per sample**
>
> SelfCheckGPT fundamentally relies on *sampling multiple continuations* and computing:
> - paragraph-level semantic divergence,
> - contradiction scores across generations,
> - n-gram overlaps vs. multiple outputs.
>
> This requires *dozens of paid API calls per query*, making it incompatible with:
> - cost-constrained settings,
> - latency-sensitive deployment,
> - models accessible only through metered APIs.
>
> More importantly, **SelfCheckGPT depends on the ability to generate alternative completions**, which is *not possible* for pre-annotated datasets such as FAVA, HaluEval, or RAGTruth, where ground truth hallucination labels correspond to **a single model response**.
> Thus, SelfCheckGPT cannot be fairly evaluated without **re-generating the entire dataset**, which is costly.
>
> ---
>
> ### **Retrieval-enhanced verifiers**
> Methods such as:
> - RARR
> - FactScore variants
> - RAG-based verifiers
>
> all require:
> - building or querying a retrieval index,
> - verifying statements using external corpora,
> - domain-specific grounding steps.
>
> These approaches evaluate *external factual consistency*, not internal calibration behaviour.
> Moreover, HUB includes **non-factual hallucinations** (logical failures, symbolic reasoning, algorithmic tasks) where retrieval-based methods do not apply.
>
> Thus, they are outside the scope of our setting.
>
> ---
>
> ## 2. Baselines That *Are* Comparable Under Our Setting
>
> For fairness, we evaluate against **baselines that operate under the same constraints**:
>
> - **white-box internal-state verifiers** such as
>   - **LLM-Check – AttnScore**
>   - **LLM-Check – HiddenScore**
> - **reference 7B hallucination classifier (FAVA model)**
> - **BlackBox BeRT Encoder (Lettuce)**
>
> These methods **do not require multiple generations**; instead, they rely on:
> - attention maps,
> - intermediate hidden states,
> - internal representations.
> - or encoder outputs.
>
> They are therefore applicable to the fixed annotated datasets.
>
> ---
>
> ## 3. Table 3: HALT vs. LLMCheck vs. FAVA
>
> On FAVA and RAGTruth—the two largest hubs within HUB—we report:
>
> - LLMCheck (AttnScore)
> - LLMCheck (HiddenScore)
> - FAVA 7B hallucination classifier
> - HALT (our 5M GRU)
> - Lettuce (BeRT base encoder)
>
> **Key finding:**
> HALT is **competitive with or surpasses** both white-box LLMCheck variants and the much larger **7B FAVA hallucination model**, despite:
> - using only log-probabilities,
> - requiring no hidden states,
> - being 1400× smaller.
>
> This strengthens our claim that hallucination signals are encoded in the **temporal structure of log-prob distributions**, and that a small GRU can learn them efficiently.
>
> ---
>
> ## 4. Summary
>
> The reviewer is correct that SelfCheckGPT and related families are important baselines in hallucination literature.
> However:
>
> ### **They cannot be evaluated on HUB because they require multiple generations per sample, which pre-annotated hallucination datasets do not provide.**
>
> ### **Our constrained setting requires single-pass, black-box-compatible methods.**
>
> Thus, we compare against:
> - **Letuce on all of HUB**
> - **LLMCheck (white-box) in FAVA and RAGTruth**
> - **FAVA 7B hallucination model in FAVA Only**
> - Lettuce (Bert base encoder)
> All of which are included in Table 3, where HALT performs competitively or better.
>
> We will clarify these methodological distinctions more explicitly in the revision.

---

### Comment · Area_Chair_MMGL · 2025-11-27
**Rebuttal and Discussion Phase**

Dear Reviewers,

Thank you again for your time and effort in reviewing this paper. We are approaching the discussion deadline. I kindly ask you to review the rebuttal and continue the discussion so that we can reach a well-considered decision.

---

### Meta-Review · Area_Chair_ULBt · 2026-01-08

**Summary:**

Reviewers acknowledged the novelty and efficiency of modeling log-probability trajectories as time series for hallucination detection and the value of the HUB benchmark.

major concerns
- (R2) the broad definition of hallucination (including reasoning errors), limited cross-model transferability, insufficient justification of design choices,
- (R1, R2) missing baselines, and weak presentation quality.
- (R3) insufficient design details of HALT and data collection process.

**Reviewer Concerns:**

addressed
- Feature attribution (gradient×input) and feature ablations showing uncertainty and rank-shift features drive HALT.
- Ablations over engineered features across HUB clusters with consistent performance drops.
- Analysis of top-k sensitivity (k = 1, 5, 10, 15, 20).
- Architectural comparison GRU vs. LSTM vs. RNN.
- Cross-model evaluation on 8 larger models (360M–70B).
- other basline comparisons.

still missing
- Clear formal taxonomy separating factual vs. logical hallucinations.
- Strong evidence that HUB advances understanding of hallucinations beyond consolidating existing datasets.

**Reviewer Scores:**

- R1, remain
- R2, remain, still negative about the contribution of the dataset and the method including feature selection
- R3, remain

---

### Decision · Program_Chairs · 2026-01-26

Reject